# Learning on the Job: An Experience-Driven, Self-Evolving Agent for Long-Horizon Tasks

## Abstract

Large Language Models have demonstrated remarkable capabilities across diverse domains, yet significant challenges persist when deploying them as AI agents for real-world long-horizon tasks. Existing LLM agents suffer from a critical limitation: they are test-time static and cannot learn from experience, lacking the ability to accumulate knowledge and continuously improve on the job. To address this challenge, we propose MUSE, a novel agent framework that introduces an experience-driven, self-evolving system centered around a hierarchical Memory Module. MUSE organizes diverse levels of experience and leverages them to plan and execute long-horizon tasks across multiple applications. After each sub-task execution, the agent autonomously reflects on its trajectory, converting the raw trajectory into structured experience and integrating it back into the Memory Module. This mechanism enables the agent to evolve beyond its static pretrained parameters, fostering continuous learning and self-evolution. We evaluate MUSE on the long-horizon productivity benchmark TAC. It achieves new SOTA performance by a significant margin using only a lightweight Gemini-2.5 Flash model. Sufficient Experiments demonstrate that as the agent autonomously accumulates experience, it exhibits increasingly superior task completion capabilities, as well as robust continuous learning and self-evolution capabilities. Moreover, the accumulated experience from MUSE exhibits strong generalization properties, enabling zero-shot improvement on new tasks. MUSE establishes a new paradigm for AI agents capable of real-world productivity task automation.

Demo videos can be found in our supplementary materials.

## 1 Introduction

In recent years, Large Language Models (LLMs) (Team et al., 2023; Yang et al., 2025; Liu et al., 2024; Dubey et al., 2024; Anthropic, 2025; OpenAI, 2025) have developed rapidly, demonstrating powerful capabilities across multiple domains. However, significant challenges remain when deploying these models as the core of AI agents designed to handle real-world tasks. While existing agents have achieved remarkable progress on standardized benchmarks such as question answering (Rein et al., 2024), mathematical reasoning (Lu et al., 2023; Mathematical Association of America, 2025), and code generation (Chen et al., 2021), these evaluations are limited to measuring domain-specific abilities. To assess the general-purpose capabilities, researchers design benchmarks in interactive environments such as OSWorld (Xie et al., 2024) and WebArena (Zhou et al., 2023). Yet, these environments still fall short, typically evaluating isolated functionalities within a single platform through short-horizon tasks of roughly 20 steps. In contrast, real-world *Productivity Tasks* represent a higher order of complexity. These tasks are characterized by long-horizon planning and interaction—potentially exceeding a hundred steps—and require agents to fluidly switch across multiple diverse applications. Such complexity demands advanced agent capabilities in long-term planning, robust interaction, and seamless cross-application tool integration.

Furthermore, most existing agents are *test-time static*. Although methods based on Reinforcement Learning (RL) can accumulate knowledge through parameter updates, they typically suffer from low sample efficiency and are restricted to training-time optimization. In the context of deploying frozen or closed-source LLMs, where parameter tuning is infeasible, their capabilities are fixed once the training phase ends. As a result, each time an agent tackles a task, it operates like an amnesiac

executor, unable to effectively learn from past experiences and lacking the capacity for continuous learning and self-evolution. Neither successes nor failures from previous tasks can be consolidated into effective knowledge to guide future actions. Consequently, even if an agent has successfully completed a task before, there is no guarantee of stable replication. When faced with repetitive tasks, it cannot improve efficiency through practice as humans do. This "one-off" interaction model severely limits agent performance in complex and dynamic environments, making efficient test-time learning difficult to implement and revealing a core deficit in the ability to truly *learn on the job without explicit weight updates*.

To address persistent challenges in dynamic planning, experience accumulation, and continuous learning for existing agents, we propose a novel agent framework **MUSE**, which stands for **M**emory-**U**tilizing and **S**elf-**E**volving. As illustrated in Figure 1, the core of MUSE is an experience-driven, closed-loop system centered around a Memory Module. This module hierarchically organizes diverse levels of knowledge, including procedural knowledge, strategic patterns, and tool-use guidance. Operating within a cross-application interactive environment, the agent leverages its accumulated experience to plan and explore solutions for long-horizon productivity tasks. After each sub-task, the agent reflects on its execution trajectory and distills reusable experience back into the Memory Module. By converting raw action sequences into structured knowledge, MUSE enhances the applicability of experience and reduces redundant exploration. This mechanism effectively extends the agent's competence beyond its static pretrained parameters, fostering

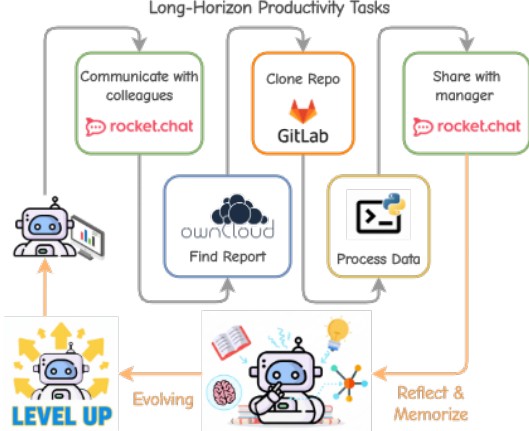

Figure 1: Illustration of the test-time learning and evolution of MUSE agents on long-horison, productivity tasks. The agent explores and accumulates experience in a cross-application interactive environment, constantly enriching its memory, thereby achieving continuous improvement.

a dynamically evolving system with superior robustness and adaptability. Crucially, since the memory is stored in natural language, the accumulated knowledge is LLM-agnostic, allowing experience gained by one model to be seamlessly transferred and utilized by another.

We evaluate our framework on TheAgentCompany (TAC) (Xu et al., 2024), a benchmark designed for long-horizon productivity tasks. Our experiments demonstrate that as the agent autonomously continuously accumulates experience within the working environment, it exhibits increasingly superior task completion capabilities and its capacity for continuous learning and self-evolution. Furthermore, our MUSE achieves new SOTA performance by a significant margin, using only a lightweight Gemini-2.5 Flash model. Our contributions are threefold:

- We present the MUSE framework, featuring an experience-driven closed-loop architecture. It empowers agents to dynamically accumulate experiences through interaction with working environments, enabling them to evolve beyond LLM's static pretrained parameters.
- MUSE autonomously converts raw action trajectories into structured, reusable memory without human intervention, aiming to reduce redundant exploration and steadily improve agent performance. Its natural language format enables seamless knowledge transfer across different LLMs.
- We establish a new SOTA on the long-horizon productivity task benchmark TAC with a score of 51.78%, achieving a 20% relative leap over the previous SOTA. Extended experiments demonstrate the effectiveness of our framework's continuous learning and self-evolution capabilities.

## 2 RELATED WORK

### 2.1 SELF-EVOLVING AGENT

The research focus in artificial intelligence is undergoing a profound paradigm shift: from developing static foundational models to building dynamic, self-evolving agents capable of continuous

adaptation and learning (Gao et al., 2025). To achieve this goal, researchers are exploring various approaches. For example, some works (Zhou et al., 2022; Wang et al., 2023b; Pryzant et al., 2023) abstract the prompt generation process into a black-box optimization problem, systematically searching for and optimizing instructions to maximize the performance of large language models (LLMs) on specific tasks. Regarding agent capability building, some cutting-edge work draws on concepts from cognitive science, helping agents accumulate skills and experience through course learning or free exploration, forming a reusable skill base (Wang et al., 2023a; Wu et al., 2024; Qian et al., 2024) or optimized toolsets (Tang et al.; Qiu et al., 2025). Another important technical approach is to empower agents with the ability to self-reflect and iterate. By introducing language feedback mechanisms and comparing and reflecting with ground truth answers, agents can continuously review and strengthen their decision-making logic and action capabilities (Shinn et al., 2023; Liang et al., 2025).

## 2.2 LLM Agent Memory Mechanisms

Research on memory mechanisms for LLM agents aims to enable them to store, retain, and recall past experiences, facilitating the transition from simple reactive models to advanced agents capable of maintaining context and autonomous adaptation. Research in this domain often draws inspiration from human cognitive models, classifying memory into short-term working memory for immediate task processing and long-term memory for persistent learning (Krishnan, 2025), which relies on external storage such as vector databases (Douze et al., 2024) and knowledge graphs (Webber, 2012). To address the challenges of information retrieval and potential information flooding arising from accumulating long-term memories, one line of research focuses on optimizing memory management and structure. For example, Mem0 (Chhikara et al., 2025) implements precise control over memory content by defining explicit memory operations, while MemInsight (Salama et al., 2025) enhances semantic information by augmenting raw memories with summaries and tags to optimize subsequent retrieval efficiency. Another branch of research focuses on constructing procedural memory by generalizing reusable experiences and workflows from agents' historical execution trajectories. Specifically, ExpeL (Zhao et al., 2024) collects execution trajectories and refines them into natural language insights and rules. Agent Workflow Memory (Wang et al., 2024b) focuses on generalizing reusable workflows from individual experiences. Memp (Fang et al., 2025) aims to build a learnable, updatable, and lifelong procedural memory, allowing agents to acquire skills and habits through experience. Although these advanced memory mechanisms are validated on various text-based benchmarks (Yang et al., 2018; Shridhar et al., 2020; Yao et al., 2022; Thorne et al., 2018) and web agent benchmarks (Zhou et al., 2023; Deng et al., 2023), existing test environments often lack sufficient complexity and long-term dependency requirements. Consequently, they may not fully assess the true efficacy of these mechanisms in handling complex, long-horizon, real-world tasks.

## 3 Methodology

### 3.1 Framework Overview

In this section, we introduce MUSE, a novel agent framework designed for *Productivity Tasks* $\mathcal{T}_{prod}$ without finetuning LLMs. To enable this test-time learning paradigm, MUSE continuously interacts with a comprehensive environment $\mathcal{E}$ that comprises multiple software and platforms, such as chat application, code editors, and web browsers. Within this environment, the agent executes actions $a_t$ via a predefined basic toolset $\mathcal{A}_{tool}$. The architecture of MUSE includes three core components designed to support this interactive learning loop: a Memory Module $\mathcal{M}$ (Sec. 3.2), a Planning-Execution (PE) Agent (Sec. 3.3), and a Reflect Agent (Sec. 3.4). The Memory Module is further decomposed into three functionally distinct components: Strategic Memory $\mathcal{M}_{strat}$, Procedural Memory $\mathcal{M}_{proc}$, and Tool Memory $\mathcal{M}_{tool}$.

As illustrated in Figure 2, the operational mechanism of our framework is a "Plan-Execute-Reflect-Memorize" iterative loop. The system begins by initializing and loading the Memory Module ($\mathcal{M}$). When a new task ($\tau \in \mathcal{T}_{prod}$) is received, the process unfolds as follows. **1) Plan and Execute:** The PE Agent initiates the process by performing a preliminary analysis of the task, decomposing it into an ordered queue of sub-tasks. For each sub-task, the PE Agent first queries the Procedural Memory to retrieve guidance from relevant prior knowledge. It then executes a sequence of actions in the interactive environment $\mathcal{E}$ using a deliberately minimal toolset. Each fundamental interaction

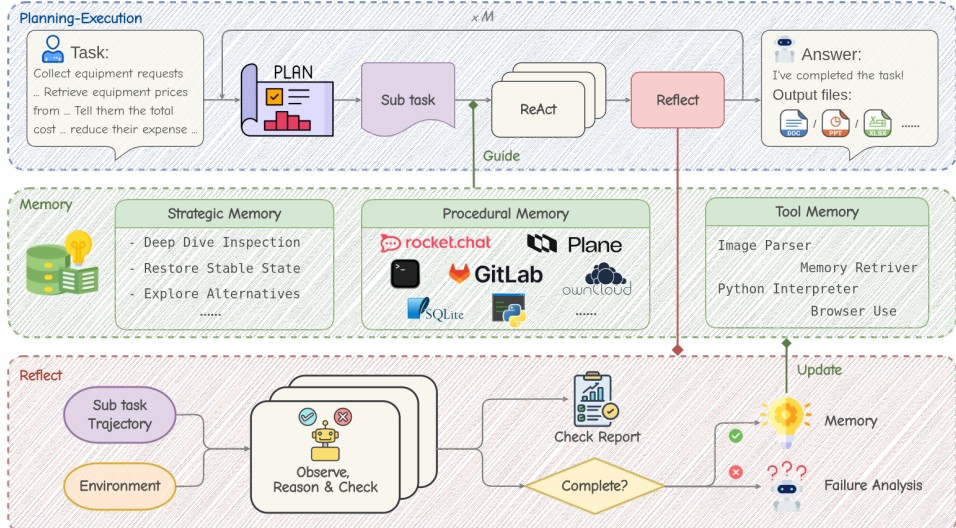

Figure 2: The MUSE framework adopts a "Plan-Execute-Reflect-Memorize" loop. The **Planning-Execution(PE) Agent** decomposes task and performs actions within an interactive environment, while the **Reflect Agent** abstracts successful attempts into Procedural Memory. After the task is completed, the Reflection Agent further synthesizes this knowledge into the Strategic and Tool Memory.

step involves the agent receiving an observation $o_t$ and, based on its history $h_t = (o_{1:t}, a_{1:t-1})$, selecting an action via its policy $a_t \sim \pi_{\text{test}}(a_t \mid h_t)$. This design compels the agent to learn how to compose primitive tool actions into complex workflows required to accomplish the sub-tasks. The execution phase for a given sub-task concludes once the PE Agent deems its attempt complete. **2) Reflect and Memorize:** After each sub-task attempt, the Reflect Agent conducts an autonomous assessment based on its environmental observations and the PE Agent's sub-task execution trajectory $h_{k:t} = (o_{k:t}, a_{k:t-1})$, requiring no human intervention. If the sub-task is successful, the Reflect Agent distills the trajectory into new *Procedural Memory*. Otherwise, it generates a diagnostic analysis of the failure and instructs the PE Agent to replan and re-execute. The PE Agent adaptively refreshes its overall task plan after each assessment, continuing this core loop until the entire task is complete. **3) Post-Task Distill:** Upon completing the overall task, a comprehensive task analysis is conducted on the full execution trajectory. From this analysis, the Reflect Agent distills higher-level *Strategic* and *Tool Memories*, capturing broader insights and effective guidelines. Ultimately, all memory types—Procedural, Strategic, and Tool—are uniformly maintained within $\mathcal{M}$, ensuring the effective retention and future applicability of all acquired knowledge.

## 3.2 MEMORY MODULE

The Memory Module $\mathcal{M}$ is the key component enabling our MUSE to learn on the job. Given the high expense of fine-tuning and our goal of maximizing the utility of closed-source LLMs, we refrain from fine-tuning the base LLM to maintain its native generalization capacity. Rather, we incrementally build up $\mathcal{M}$, allowing the agent's performance $R(t)$ to improve over repeated trials on tasks $\tau \in \mathcal{T}_{prod}$. This module is a composite memory $\mathcal{M} = \{\mathcal{M}_{strat}, \mathcal{M}_{proc}, \mathcal{M}_{tool}\}$ comprising three distinct memory types, each optimized for a specific level of abstraction: Strategic Memory $\mathcal{M}_{strat}$ for macro-level behavioral paradigms, Procedural Memory $\mathcal{M}_{proc}$ for combinations of tool sequences, and Tool Memory $\mathcal{M}_{tool}$ for individual tool use. Each memory type operates with distinct mechanisms for its generation, updating, and application.

**Strategic Memory** ($\mathcal{M}_{strat}$) focuses on distilling lessons from dilemmas an agent encounters during task execution and their solutions, particularly from challenges that require multiple attempts to overcome. The Reflect Agent abstracts these "problem-solution" experiences into high-level guidance and formats them as <Dilemma, Strategy> key-value pairs. Upon agent initialization, the entire $\mathcal{M}_{strat}$ is loaded into the system prompt to guide its global task execution strategy. To ensure efficiency and prevent context window bloat, this memory is updated, merged, and refined after each task, always maintaining a concise size. For specific examples, refer to Table 6 in the appendix.

**Procedural Memory** ($\mathcal{M}_{proc}$) archives the PE Agent's successful sub-task trajectories as a hierarchical knowledge base of Standard Operating Procedures (SOPs). This library is indexed first by application (e.g., related platforms or APIs), followed by a second-level SOP index that documents the key analyses, precaution, core parameters, and operational steps for each sub-task. To balance efficiency and performance, the system employs a lightweight, proactive retrieval mechanism. Only the SOP index is loaded at startup to minimize overhead. When facing uncertainty, the agent utilizes a built-in tool to proactively query detailed SOPs for decision support, which closely mimics how human experts consult past cases. The memory system is refined through a two-stage process. First, immediately following a successful sub-task, the Reflect Agent dynamically adds the new SOP $p_{new}$ to $\mathcal{M}_{proc}$ for immediate reuse. Second, after the entire task is complete, the agent performs a higher-level, global refinement (e.g., deduplication, generalization) to continuously optimize the long-term quality and applicability of the knowledge base. See Table 7 in the appendix for examples.

**Tool Memory** ($\mathcal{M}_{tool}$) functions as the agent's "muscle memory" for single tool usage, operating automatically without requiring proactive retrieval. This memory consists of two components, $\mathcal{M}_{tool} = \{D_{static}, I_{dynamic}\}$: A Static Description $D_{static}$, loaded into the system prompt at startup to explain each tool's core functionality, and a Dynamic Instruction $I_{dynamic}$, which is returned with the environment's observation $o_t$ after a tool is used. This instruction guides the agent's immediate next action $a_{t+1}$, such as suggesting a subsequent tool to invoke or an analysis to perform. To ensure this "muscle memory" improves over time, the Tool Memory is updated by the Reflect Agent after each task is completed. See Table 8 in the appendix for specific examples.

### 3.3 PLANNING-EXECUTION AGENT

Productivity tasks often require dozens of coordinating actions across multiple applications. To manage this complexity, the PE Agent first decomposes the main task $\tau$ into an ordered queue of sub-tasks $Q = [st_1, st_2, \ldots, st_M]$ based on the initial task description. The agent then systematically works through this queue, attempting to resolve each sub-task $st_i$ via an iterative ReAct (Yao et al., 2023) process. Crucially, after each sub-task execution, the agent re-evaluates and updates the sub-task queue $Q$ based on newly acquired information, ensuring an adaptive path to task completion.

**Sub-task Plan and Replan.** Both initial planning and subsequent replanning follow a unified, multi-turn process that generates an ordered sub-task queue $Q$. Each sub-task $st_i \in Q$ is defined by a tuple $st_i = (\text{desc}_i, \text{goal}_i)$, where $\text{desc}_i$ outlines its scope and $\text{goal}_i$ serves as the evaluation basis for the Reflect Agent. The primary distinction between the two phases lies in their inputs. The initial plan $Q_{init}$ is derived solely from the user's original task description. In contrast, replanning is a dynamic process that occurs after each sub-task is attempted. It integrates the execution results and the Reflect Agent's assessment to continuously refine the current plan. When $Q$ is empty, the PE Agent performs a final review, examining the global state of the environment to confirm that the overall task objectives have been met. By iteratively maintaining and updating $Q$, MUSE ensures the stable and coherent execution of long-horizon tasks and prevents error accumulation.

**Sub-task Execute and Retry.** The PE Agent processes sub-tasks $st_i$ sequentially from the queue $Q$, attempting to resolve each one using a memory-enhanced ReAct loop. The core of this loop is the iteration of a $(\theta_t, a_t, o_t)$ tuple, representing Thought, Action, and Observation: the agent first generates a thought to plan an action $a_t$—such as entering text, clicking a button, or querying its Procedural Memory $\mathcal{M}_{proc}$—then executes the action and receives an observation $o_t$ as feedback. This cycle continues until the agent concludes that the sub-task's goal has been met. To prevent the agent from getting stuck in futile loops, a maximum of $N$ actions is imposed on each sub-task attempt. If this limit is reached, the Reflect Agent intervenes to evaluate and grant one retry opportunity. This retry mechanism is explicitly designed to encourage exploration over exploitation. During the retry, the PE Agent is no longer required to use Procedural Memory $\mathcal{M}_{proc}$, enabling it to discover novel methods when existing knowledge is erroneous or inapplicable. If this second attempt also fails, the PE Agent triggers a sub-task replanning process.

**Minimal Usable Toolset.** In contrast to many general agent studies (Qin et al., 2023; Patil et al., 2024) that aim to integrate a massive number of APIs, we equip MUSE not with specialized tools for specific applications (like PDF or Excel), but with a minimal toolset $\mathcal{A}_{tool}$ of fundamental yet powerful general-purpose tools. This toolset includes browser interaction, a code interpreter, a Shell, a vision extractor, and a memory retriever. We believe that the core of intelligence lies in the ability to creatively combine basic tools, rather than mechanically invoking pre-defined functions. Further-

more, a key objective of this research is to validate whether MUSE can convert successful solutions into reusable Procedural Memory, thereby achieving the self-evolution of its capabilities. A full list of our toolset are illustrated in Appendix A.4 and Table 9.

**Procedural Memory Retrieval.** To achieve low-cost experience reuse while respecting the LLM's context length limit, the experience retrieval mechanism separates the memory index from the detailed content. An SOP $p \in \mathcal{M}_{proc}$ is thus structured as a pair $p = (index_p, content_p)$. At the start of a sub-task, only a lightweight index of all available SOPs, $I_{\mathcal{M}_{proc}} = \{index_p \mid p \in \mathcal{M}_{proc}\}$, is loaded into the context. The PE Agent can then, at any point during execution, use a dedicated tool $a_{mem}$ to retrieve the full $content_p$ of a specific SOP on demand. To maximize the value of this feature, we use prompt engineering to encourage the agent to prioritize querying for relevant experience at the beginning of each sub-task.

### 3.4 Reflect Agent

During execution, the PE Agent can encounter hallucinations (e.g., erroneously believing a task is complete) and failures. To address this, the Reflect Agent acts as an independent, third-party supervisor. For its analysis, it receives the sub-task's definition $st_i = (\text{desc}_i, \text{goal}_i)$, along with the PE Agent's sub-task execution trajectory $h_{k:t}$. Notably, it can also interact directly with the environment $\mathcal{E}$ to independently verify information.

**Sub-task Evaluation.** The Reflect Agent's evaluation process is triggered whenever the PE Agent completes a sub-task or reaches its action limit $N$. It starts by formulating an ordered checklist based on three core dimensions: 1) Truthfulness Verification: Ensuring conclusions are grounded in real environmental feedback to suppress hallucinations. 2) Deliverable Verification: Checking the existence, completeness, and correctness of any output files or reports. 3) Data Fidelity: Confirming that data has not been lost, truncated, or altered during processing.

To execute this checklist, the Reflect Agent, which is equipped with the same toolset $\mathcal{A}_{tool}$ as the PE Agent, utilizes two primary inspection methods. The first is *trajectory referencing*, which explicitly traces the PE Agent's conclusions back to specific observations $o_t$ in the execution history $h_{k:t}$. The second is *active verification*, which involves proactively using tools to interact with the environment $\mathcal{E}$ and cross-check key information with real-time feedback.

Upon completing its checks, the Reflect Agent outputs a success/failure flag $f$ and a detailed check report. This tuple is fed back to the PE Agent as a historical record. Based on the outcome, the Reflect Agent then performs a critical operation: if $f = $ success, it summarizes the effective operational sequence from $h_{k:t}$ into a new SOP $p_{new}$ for the Procedural Memory $\mathcal{M}_{proc}$; if $f = $ failure, it generates a failure cause analysis report $R_{fail}$. Finally, based on this complete evaluation, the PE Agent initiates the necessary replanning.

**Memory Update Mechanism.** The entire task $\tau$ is considered complete once the PE Agent stops generating new subtasks in the replanning phase. The PE Agent then launches a task review, summarizing its execution attempts and outcomes. This triggers the Reflect Agent to conduct a full-scale upgrade of the memory system $\mathcal{M}$. It begins by analyzing task challenges and solutions to extract <Dilemma, Resolution Pattern> pairs, thereby reinforcing Strategic Memory $\mathcal{M}_{strat}$, while also codifying effective tool usage to augment Tool Memory $\mathcal{M}_{tool}$. Then, all three types of memory undergo a thorough refinement and integration process, aiming to integrate new and old knowledge, eliminate redundancy, and generalize common patterns within $\mathcal{M}$.

## 4 Experiments

### 4.1 Benchmark

Our experimental evaluation utilizes TheAgentCompany (TAC) benchmark (Xu et al., 2024). Comprising 175 tasks, this benchmark is designed to assess the comprehensive capabilities of autonomous language agents by simulating a high-fidelity corporate environment. The tasks are structured around six core employee positions (e.g., HR, PM, SDE), requiring the agent to execute interconnected operations using a suite of applications such as chat clients, cloud storage, and project management software, all within a fully functional operating system. A core feature of TAC is

the high complexity and long-horizon nature of its tasks. On average, completing a task requires over 40 action steps, frequently spanning two or more applications. This demands that an agent decompose high-level objectives into a coherent, protracted sequence of steps and integrate information across platforms. Therefore, TAC provides a rigorous platform for evaluating an agent's real-world problem-solving, multi-step planning, and long-horizon reasoning capabilities, making it highly suitable for our research focus on long-horizon productivity tasks.

## 4.2 EXPERIMENTAL SETUP

In our experimental configuration, the PE Agent and Reflect Agent employ the Gemini-2.5 Flash model (Comanici et al., 2025), while NPCs in the TAC environment are powered by GPT-4o model. The maximum number of actions for each sub-task is set to $N = 20$. For evaluation, we rely on the official protocol provided by the TAC benchmark. This evaluation protocol not only assesses the final completion status of a task but also defines a series of critical intermediate checkpoints to measure partial progress. The primary metric is the partial completion score ($S_{partial}$), which is calculated as: $S_{partial} = 0.5 \cdot \text{Completed\_ckpt}/\text{Total\_ckpt} + 0.5 \cdot S_{full}$, where $S_{full} \in \{0, 1\}$ is a binary indicator of whether the task was fully completed. Our final performance metric is computed as the average partial completion score across all evaluated tasks. Additionally, we calculate an aggregate checkpoint score, $S_{ckpt}$, which represents the proportion of completed checkpoints relative to the total checkpoints across all tasks.

## 4.3 EXPERIMENTAL RESULTS

### 4.3.1 CONTINUOUS LEARNING EXPERIMENTS

To validate the continual learning capability of our MUSE framework, we curated a subset of 18 tasks from TAC benchmark, which we denote as $\mathcal{T}_{cl}$. This subset was sampled to ensure coverage across all six professional roles. Our experimental design simulates how humans accumulate experience, with the primary objective of testing whether the agent can progressively improve its performance on repetitive tasks by continuously updating its memory. For comparison, we first establish a baseline by evaluating the Gemini-2.5 Flash model on $\mathcal{T}_{cl}$ without our Memory Module. The main experiment then consists of three sequential iterations with no human intervention, where in each iteration, the agent tackles all 18 tasks in order, carrying its accumulated knowledge forward to the next. To mitigate randomness, we conduct five complete runs of this experiment and report the average scores. The detailed tasks in $\mathcal{T}_{cl}$ are elaborated in Appendix A.1 and Table 5.

As illustrated in Figure 3, the results clearly show that both the $S_{ckpt}$ and $S_{partial}$ metrics grew steadily and monotonically across the three iterations, a direct manifestation of our framework's self-evolving capability. Most critically, in the final round, MUSE outperformed the memory-less baseline by over 10%, confirming the effective translation of self-accumulated knowledge into substantial performance gains. We attribute this advantage to the accumulated experience, which enables the agent to avoid previously failed exploration paths and thus focus more directly on effective solutions. This approach not only improves efficiency by streamlining the LLM's context but also enables the agent to achieve an unprecedented depth of exploration.

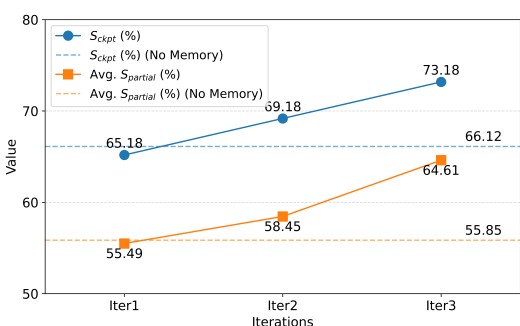

Figure 3: Performance trends across iterations of MUSE. The figure shows $S_{ckpt}$ (blue) and average $S_{partial}$ (orange). Dashed lines denote the baseline without memory, while solid lines track improvements across iterations.

### 4.3.2 GENERALIZATION EXPERIMENTS

To further evaluate the generalization capability of our memory mechanism, we curate a challenging subset from the TAC benchmark, denoted as $\mathcal{T}_{hard}$. This subset comprises 12 tasks on which even strong models like Claude-4 Sonnet achieve little to no score. The purpose of this experiment is to

Table 1: Performance comparison on the hard-task set. The agent with memory demonstrates significant improvement.

| Framework | Model | checkpoint | $S_{ckpt}$ (%) ↑ | Avg. $S_{partial}$ (%) ↑ |
|---|---|---|---|---|
| Openhands-versa (Soni et al., 2025) | claude-4 sonnet | 3 / 59 | 5.08 | 2.00 |
| Openhands (Wang et al., 2024a) | gemini-2.5 pro | 5 / 59 | 8.47 | 3.00 |
| MUSE w/o mem | gemini-2.5 flash | 18 / 59 | 30.51 | 23.65 |
| MUSE w/ mem | gemini-2.5 flash | **24 / 59** | **40.68** | **33.41** |

test our agent's zero-shot generalization ability when facing entirely new and highly difficult tasks. Specifically, we compare a baseline agent operating without memory against our agent equipped with the fixed Memory Module, which is accumulated over three iterations of continual learning on the $\mathcal{T}_{cl}$ set. Both agents are evaluated on this previously unseen $\mathcal{T}_{hard}$ task set. By comparing their performance differences, we can easily determine whether the memory mechanism can effectively transfer historical experience to unknown scenarios. Detailed $\mathcal{T}_{hard}$ are elaborated in Appendix A.1.

As shown in Table 1, current SOTA agents, such as the Openhands framework (Wang et al., 2024a) using powerful closed-source models like Gemini-2.5 Pro (Comanici et al., 2025) and Claude-4 Sonnet (Anthropic, 2025), struggle significantly on $\mathcal{T}_{hard}$. In general, they complete fewer than 10% of the checkpoints and achieve an $S_{partial}$ of only 2% or 3%. In contrast, our MUSE, using only the lighter Gemini-2.5 Flash model, reaches an $S_{partial}$ of 23.65% even without relying on memory, which demonstrates the effectiveness of the synergy between our PE and Reflect Agents. When equipped with its pre-learned memory, our agent's performance further improves to a remarkable $S_{partial}$ of 33.41%. This result provides strong evidence for the zero-shot generalization capability of the knowledge acquired through our framework, indicating that MUSE learns transferable and generalizable memory, rather than merely remembering task-specific solutions.

### 4.3.3 TAC Full Benchmark

To conduct a fair and comprehensive comparison against other leading agent methods, we evaluated our framework on the complete TAC benchmark, which includes all 175 tasks. For this experiment, the agent was equipped with the Memory Module accumulated after three iterations on the $\mathcal{T}_{cl}$ subset, and this memory was kept frozen throughout the evaluation.

As shown in Table 2, MUSE achieves new SOTA performance on the TAC benchmark. Notably, it attains an average $S_{partial}$ of 51.78%, marking the first time an agent has surpassed the 50% threshold on this benchmark and outperforming the previous SOTA (OpenHands-versa w/ Claude-4 Sonnet) by nearly 20%. This substantial improvement is particularly striking given that our agent's memory was acquired from only approximately 10% of the available tasks, demonstrating exceptional "learning on the job" efficiency. These results prove the effectiveness of MUSE in challenging real-world productivity tasks. Besides, they provide strong empirical support for our core thesis: past condensed experience yields highly generalizable capabilities that far exceed what might be expected from the limited learning. The complete results of 175 tasks are listed in Table 10.

Table 2: Performance comparison across frameworks and models on the TAC full 175-task benchmark. **PCR** (Perfect Completion Rate) indicates the proportion of tasks that are fully solved.

| Framework | Model | checkpoint | $S_{ckpt}$(%) ↑ | Avg. $S_{partial}$(%) ↑ | PCR(%) ↑ |
|---|---|---|---|---|---|
| OWL-RolePlay (Hu et al., 2025) | gpt-4o + o3-mini | 127 / 776 | 16.37 | 11.04 | 4.00 |
| Openhands (Wang et al., 2024a) | gemini-1.5 pro | 90 / 776 | 11.60 | 8.02 | 3.43 |
| | gemini-2.0 flash | 195 / 776 | 25.13 | 18.96 | 11.43 |
| | gemini-2.5 pro | 361 / 776 | 46.52 | 39.28 | 30.29 |
| Openhands-Versa (Soni et al., 2025) | claude-3.7 sonnet | 353 / 776 | 45.49 | 40.18 | 30.86 |
| | claude-4 sonnet | 392 / 776 | 50.52 | 43.19 | 33.14 |
| MUSE (ours) | gemini-2.5 flash | **465 / 776** | **59.92** | **51.78** | **41.14** |

## 4.4 ABLATION STUDY

### 4.4.1 ABLATION STUDY FOR REFLECT AGENT

To evaluate the impact of the Reflect Agent in the MUSE framework, we conduct an ablation study by removing it. Specifically, we compare a variant lacking the Reflect Agent against the full framework, with both configurations operating without the Memory Module. As presented in Table 3, the non-reflective variant underperformed on the 18-task subset $\mathcal{T}_{cl}$. This result demonstrates the indispensable role of the reflective mechanism in ensuring execution quality and providing the high-quality signal required for effective learning.

Table 3: Performance comparison of ablation studies of Reflect Agent on $\mathcal{T}_{cl}$. Results show that removing the Reflect Agent (*No Reflection Variant*) leads to a substantial performance drop.

| Framework | Model | checkpoint | $S_{ckpt}$ (%) ↑ | Avg. $S_{partial}$ (%) ↑ |
|---|---|---|---|---|
| No Reflection Variant | gemini-2.5 flash | 54 / 85 | 63.53 | 43.21 |
| MUSE | gemini-2.5 flash | **56.2 / 85** | **66.12** | **55.85** |

### 4.4.2 ABLATION STUDY FOR DIFFERENT MODELS

To evaluate the open-source LLM adaptability of our framework, we replace the core model with DeepSeek-V3-250324 (Liu et al., 2024) and conduct experiments in two scenarios: with and without the pre-accumulated memory on the 18-task subset $\mathcal{T}_{cl}$. The results, when compared against other open-source-based agents in Table 4, yield two key insights. First, even without memory, the MUSE architecture alone enables DeepSeek-V3 to outperform all other frameworks using open-source models, highlighting the intrinsic advantages of our design. Second, the addition of the pre-accumulated Memory Module provides a significant performance boost, confirming that our memory mechanism is model-agnostic and that the accumulated knowledge can be effectively transferred across different LLMs.

Table 4: Performance comparison of agents utilizing open-source LLMs on $\mathcal{T}_{cl}$.

| Framework | Model | checkpoint | $S_{ckpt}$ (%) | Avg. $S_{partial}$ (%) |
|---|---|---|---|---|
| Openhands | llama-3.1 405b | 17 / 85 | 20.00 | 9.78 |
| | llama-3.3 70b | 11 / 85 | 12.94 | 5.84 |
| | qwen-2.5 72b | 11 / 85 | 12.94 | 6.50 |
| MUSE w/o memory | deepseek-v3 | 29 / 85 | 34.12 | 28.01 |
| MUSE w/ memory | deepseek-v3 | **43 / 85** | **50.59** | **36.75** |

## 5 DISCUSSIONS AND CONCLUSION

**Discussions.** We employ memory modules to tackle long-horizon productivity tasks (some spanning over 100 steps), as fine-tuning methods suffer from computational intractability, while RL-based approaches are hindered by the design of rewards that are both extremely sparse and difficult to formulate. Thus, this research focuses on enhancing agent memory to empower test-time learning capabilities. We acknowledge that our current memory architecture is not a panacea and has limitations in handling specific tasks like high-level planning or multi-hop search. Nevertheless, the experimental results confirm its potential. We attribute this success to the agent's ability to efficiently avoid previously failed paths and reallocate exploration to more promising regions, effectively pruning the decision space and enabling a deeper, more successful search.

The TAC benchmark represents a significant step forward in evaluating agents on complex tasks, which is a key reason we selected it to test our framework. However, during our experiments, we also observed some limitations. Some task descriptions can be ambiguous or contain inaccuracies. Furthermore, the evaluation scripts for certain tasks are rigid and do not account for the full range of valid solutions. As a result, several unexpected yet plausible agent strategies are sometimes underestimated or incorrectly penalized. We provide two detailed case studies illustrating these issues in Appendix A.2.

**Conclusion.** In this study, we propose an experience-driven self-evolving framework, MUSE. Centered around a hierarchical Memory Module, MUSE systematically extracts reusable knowledge from interaction trajectories to tackle complex, long-horizon productivity tasks. Comprehensive evaluation on the TAC benchmark confirms MUSE's effectiveness: it achieves continuous performance improvement and self-evolution with autonomous experience accumulation, and shows remarkable experience generalization to novel tasks. Consequently, MUSE achieves a new SOTA performance on TAC by a large margin.

## ETHICS STATEMENT

This work does not involve human subjects, sensitive data, or applications with foreseeable ethical concerns. We have carefully reviewed the ICLR Code of Ethics and confirm that our study complies fully with its requirements.

## REPRODUCIBILITY STATEMENT

To ensure the reproducibility of our work, we have made every effort to provide all necessary details and materials. The complete experimental setup, including hyperparameters, model configurations, and evaluation protocols, is described in detail in Section 4.2 of the main text. In addition, we present specific examples of the MUSE framework's Memory Module in Appendix A.3, and we comprehensively report the experimental results on all 175 tasks of TheAgentCompany benchmark in Table 10 in the appendix.

All experiments were conducted using publicly available models and the publicly released TheAgentCompany benchmark. Our framework is designed to be directly re-implementable, and we commit to releasing the full codebase, experimental logs, configuration files, and evaluation scripts upon acceptance of this paper, in order to facilitate future research and ensure complete reproducibility of our results.

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

# A APPENDIX

## A.1 SELECTED TASK SPLITS

In constructing the task splits, we followed clear selection criteria. For the continuous learning task set $\mathcal{T}_{cl}$, we chose tasks of moderate difficulty, specifically those where models generally achieve non-zero scores in the official TAC evaluation, ensuring that they are neither trivial nor impossible. For the $\mathcal{T}_{hard}$ task set, we deliberately selected tasks where most models fail almost completely, i.e., tasks that typically yield near-zero scores, to better stress-test the limits of the agents. In both cases, we ensured a comprehensive coverage across the six professional domains defined in TAC, and every task was manually inspected to guarantee correctness and to exclude cases with obvious errors. We list these tasks in detail in Table 5.

Table 5: Task Sets Overview

| Set Name | Task Name |
|---|---|
| $\mathcal{T}_{cl}$ (18) | admin-check-employees-budget-and-reply-and-record |
| | admin-read-survey-and-summarise |
| | ds-sql-exercise |
| | ds-answer-spreadsheet-questions |
| | ds-visualize-data-in-pie-and-bar-chart |
| | finance-check-attendance-payroll |
| | finance-budget-variance |
| | hr-collect-feedbacks |
| | hr-new-grad-job-description-3 |
| | hr-transfer-group |
| | hr-check-attendance-multiple-days-department-with-chat |
| | pm-create-channel-message-medium |
| | pm-update-plane-issue-from-gitlab-status |
| | pm-ask-for-issue-and-create-in-gitlab |
| | pm-check-backlog-update-issues |
| | sde-update-dev-document |
| | sde-update-issue-status-on-plane |
| | sde-add-all-repos-to-docs |
| $\mathcal{T}_{hard}$ (12) | admin-mass-forms-filling |
| | ds-calculate-spreadsheet-stats |
| | ds-predictive-modeling |
| | finance-invoice-matching |
| | finance-nonqualified-bill-ask-for-reimburse |
| | hr-mass-survey |
| | hr-internal-tooling-slides |
| | hr-salary-analysis |
| | pm-present-engineer-group-members |
| | sde-copy-table-from-pdf-to-xlsx |
| | sde-sotopia-create-agent-wo-repo |
| | sde-create-commit-table-for-all-gitlab-users |

## A.2 CASE STUDIES

We will first present two representative case studies that illustrate both the inherent complexity of TAC tasks and the operating mechanisms of our framework. These cases reveal unexpected yet effective completion paths that highlight the adaptive problem-solving capabilities of our agent.

Figure 4 shows the first case study, which involves a task requiring the agent to collect performance feedback on Liu Qiang from three colleagues. The standard approach—and the one implicit in TAC evaluation protocols—would involve conducting sequential individual conversations with each colleague. Nevertheless, our agent adopted an alternative strategy during the planning phase and achieved task completion, yet this innovative solution fell outside the scope of TAC evaluation protocol. Specifically, it created a multi-person chat group and simultaneously queried all three colleagues. This innovative approach significantly streamlined the information collection process. Ultimately, the agent accurately integrated the feedback into a comprehensive performance evaluation for Liu Qiang, demonstrating its efficiency and adaptability in task execution.

The second case study presented in Figure 5 demonstrates the agent's sophisticated capabilities in processing dynamically acquired information and executing complex, long-horizon cross-platform tasks. In this scenario, the agent orchestrated nearly 20 sub-tasks and performed over 100 actions. The primary objective was to create a new issue in the RisingWave project on GitLab. However, the task description also implied that the task would be assigned to an engineer named Li Ming. This overloaded task was obviously not considered by the TAC, because they did not set up a GitLab account for Li Ming. The task posed two major challenges: 1) To gather the comprehensive details necessary for issue creation, the agent needed to sequentially consult three different colleagues. However, the initial task provided only a single contact person (Li Ming). The agent had to progressively identify and locate the other two relevant colleagues through interactions with Li Ming, while continuously adapting and refining its sub-task queue in real-time. 2) The agent attempted to assign the issue to Li Ming but could not find his GitLab account, which was an important prerequisite. The agent discovered the problem and decided to create a GitLab account for Li Ming. The agent then performed a series of actions, including creating the account and adding Li Ming to the appropriate project team, ultimately successfully assigning the issue to him.

Despite the inherent challenges and profound complexity of the task, the agent demonstrates a remarkable capacity for autonomous adaptation. Through dynamic sub-task planning, self-reflection, and continuous optimization via trial and error, it not only achieves but even exceeds its original task objectives.

**Failure cases study.** Here, we analyze the failure mechanisms observed in the TAC tasks and present two specific cases.

A critical contributing factor is the inherent limitation of environmental observation. As illustrated in Figure 6, the agent failed in its processing flow due to a lack of preemptive perception of unstructured data, such as images embedded within PDFs. Similarly, Figure 7 demonstrates that insufficient observation of structured data formats (e.g., thousands separators in Excel) resulted in the inability to extract valid numerical values during the code generation and execution phases.

These failure scenarios underscore a fundamental disparity between the perception modalities of agents and humans: humans possess active intuition and parallel multi-modal perception, enabling the instantaneous construction of global cognition. In stark contrast, the agent's perception remains passive and linear. Lacking prior environmental intuition, the agent frequently encounters blind spots when handling non-standard data or complex environments.

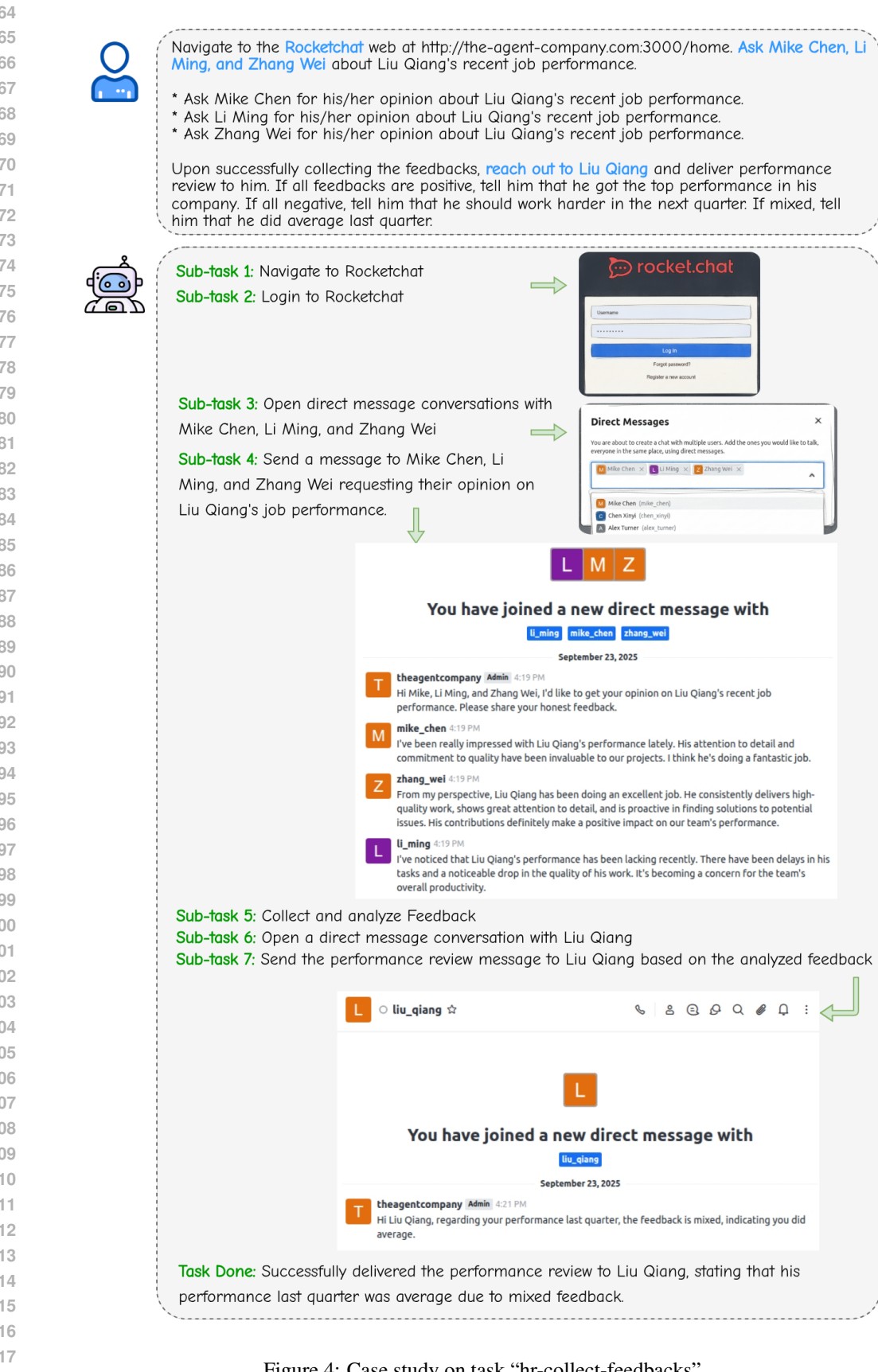

Figure 4: Case study on task "hr-collect-feedbacks"

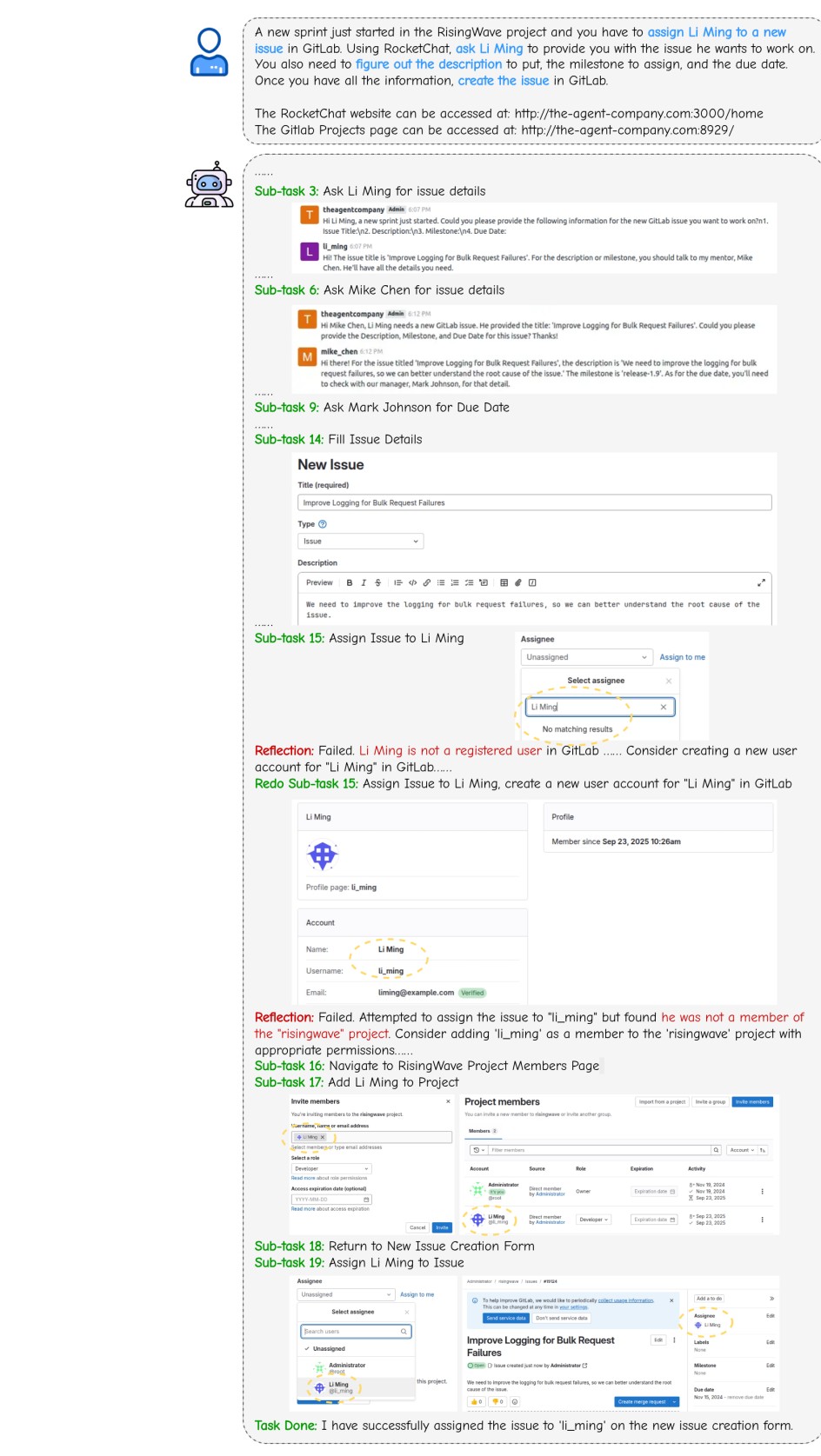

Figure 5: Case study on task "pm-ask-for-issue-and-create-in-gitlab"

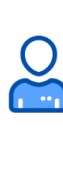

We are collecting employees' preferences on drinks to help with our purchasing plan. Please navigate to http://the-agent-company.com:8092/ and find drinks_survey.pdf, which contains a questionaire that we have placed in the office. Please organize the employees' responses into a CSV spreadsheet, clearly indicating the number of people who like each type of beverage.

The spreadsheet is prepared for you at /workspace/drinks_survey.csv and please complete it.

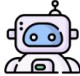

......

**Sub-task 3:** Extract text from drinks_survey.pdf

**Action 1:** Access_the_application_guide

......

**Action 3:** CMD：Convert PDF to txt file

<tool_call>
name: run_cmd
args:
    cmd_str: "pdftotext /workspace/drinks_survey.pdf \
/workspace/drinks_survey.txt"
</tool_call >
> returncode: 0

**Action 4:** CMD：Verify that the txt file is generated

<tool_call>
name: run_cmd
args:
    cmd_str: "ls -l /workspace/drinks_survey.txt"
</tool_call >
> -rw-r--r-- 1 root root 266491 Sep 14 09:58 /workspace \
/drinks_survey.txt

**Sub-task 4:** Read text content

**Action 1:** CMD：Read drinks_survey.txt

<tool_call>
name: run_cmd
args:
    cmd_str: "cat /workspace/drinks_survey.txt"
</tool_call >
> The Drinks You Want Us to Buy
Coke
Sprite
Mountain Dew
Apple Juice
Orange Juice

**Note:**
The agent failed to extract all relevant data because it did not account for images in the PDF, such as charts or graphs. These images, which likely contained critical information, were missed during the text extraction process. As a result, valid data present in these images was not captured, leading to incomplete or inaccurate results.

**Action 2:** I read the text from the PDF, but it seems there are no clear statistical results visible in the extracted content......

......

The agent generated incorrect results due to failing to account for images in the PDF, which led to valid data (such as charts or graphs) being missed during the text extraction process.

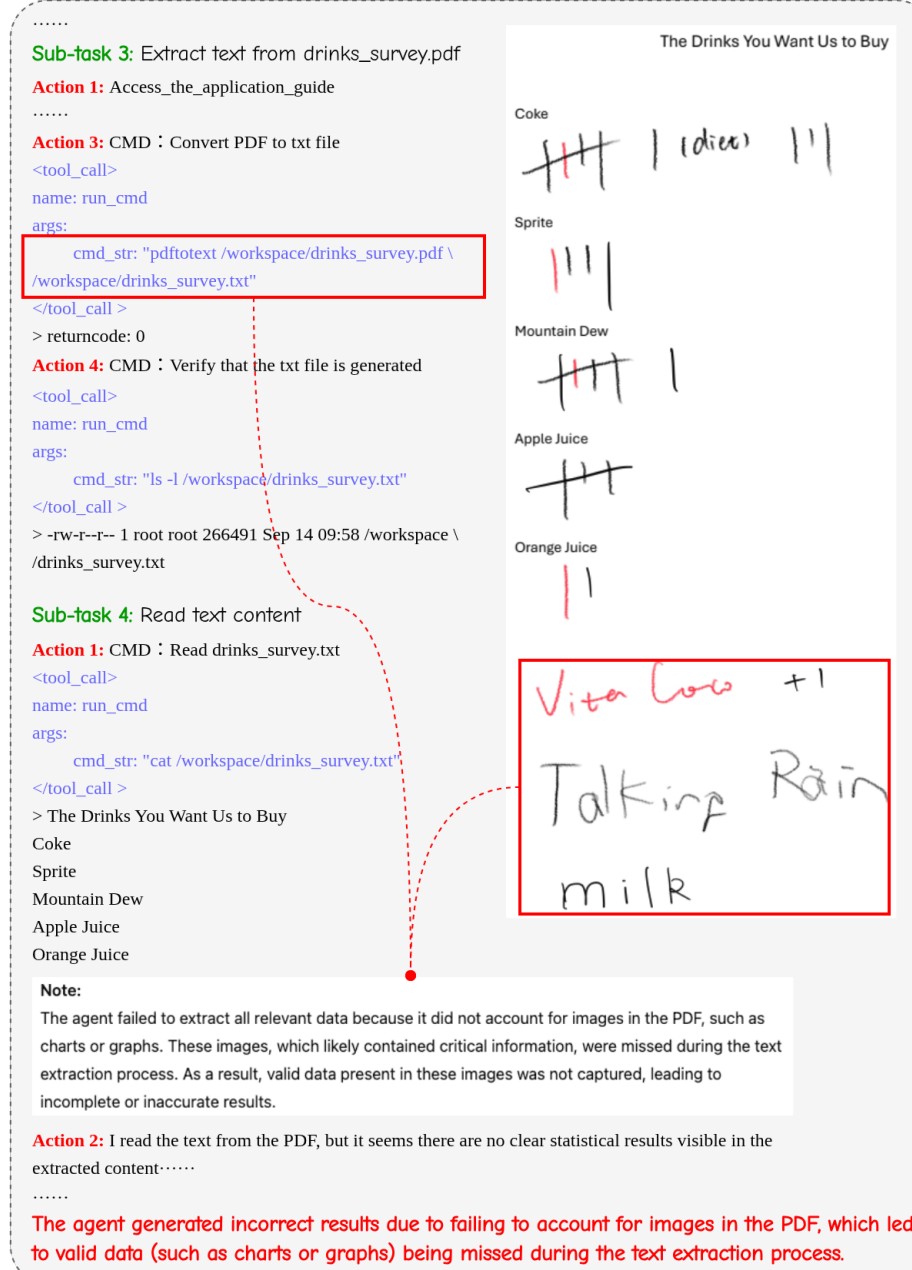

Figure 6: Case study on task "admin-make-spreadsheet"

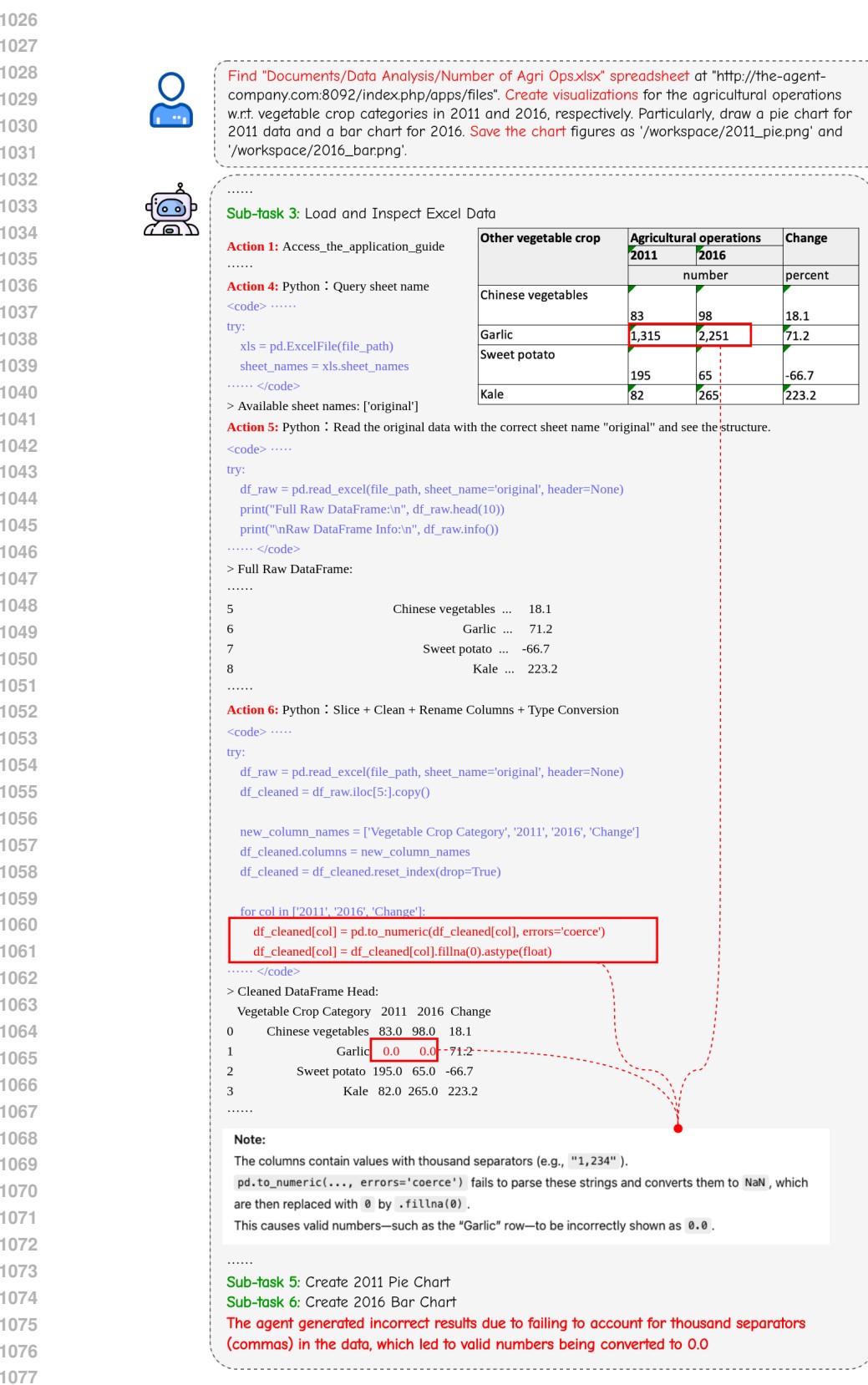

Figure 7: Case study on task "ds-visualize-data-in-pie-and-bar-chart"

## A.3 EXEMPLARY DEMONSTRATION OF THE MEMORY MODULE

In this section, we provide a concrete illustration of the three memory types in our framework. We present selected entries from the Strategic Memory (Table 6), Procedural Memory (Table 7), and Tool Memory (Table 8) to demonstrate their structure and content. These examples are shown in their native, structured format.

Table 6: Examples of $\mathcal{M}_{strat}$, outlining high-level principles for robust agent behavior.

| Principle | Description |
|---|---|
| Systemic Root Cause | Diagnose and address underlying systemic causes of recurring errors to refine methods and ensure long-term stability beyond symptom treatment. |
| Robust Context State | Explicitly manage and continuously verify data and execution context throughout its lifecycle to ensure accuracy, consistency, and integrity of dependencies and prevent errors. |
| Adaptive Task Progression | Implement primary strategies with adaptive fallbacks and dynamically provision prerequisites, ensuring continuous progression and reliable state transitions even when initial paths are blocked. |
| Problem Decomposition | Decompose complex problems into modular, manageable units, defining clear goals and objectives to structure a stable and logical execution path. |
| Granular Outcome Verification | After critical state-changing actions, perform detailed, item-by-item verification of all intended outcomes to detect subtle discrepancies and ensure system state precisely matches requirements. |
| Iterative Data Extraction | Employ adaptive search heuristics and staged parsing strategies, including resilient capture mechanisms, to reliably extract, validate, and process information from dynamic or complex data sources. |
| Accurate Output Specification | Employ flexible methods or custom logic to achieve exact output specifications, ensuring the final representation precisely matches requirements, intent, and diverse formats. |
| Explicit Uncertainty Handle | When required information is unextractable or unverifiable, explicitly assign a clear "Not Available" or equivalent status to prevent hallucination and maintain data integrity. |
| Clear Environment Separate | Strictly distinguish and manage execution environments for different types of code or tools (e.g., shell vs. Python) to prevent conflicts and ensure proper, intended execution. |

Table 7: Examples of $\mathcal{M}_{proc}$, detailing step-by-step guides for common application interactions.

| Application | Function | Details |
|---|---|---|
| RocketChat | Navigate to Home Page | **Preconditions:** User is logged into RocketChat. (Optional: Browser is open). **Steps:** Refresh the page state using `browser_update` → If already at `/home`, verify. Otherwise, click the 'Home' link or navigate directly to `http://xxx/home`, refresh and verify elements like 'Home' button, 'Channels' list, or avatar. **Notes:** Always follow navigation with `browser_update`. The 'Home' link may be more reliable than a button depending on UI context. |
| | Login | **Preconditions:** Browser is open, RocketChat URL and credentials are known. **Steps:** Navigate to login page → Enter username in 'Email or username' field → Enter password in 'Password' field → Click 'Login' button → Verify login success (URL is `/home`, login fields disappear, post-login elements appear). **Notes:** Successful login is confirmed by disappearance of input fields and presence of post-login UI. Always perform `browser_update` after clicks. |
| FileSystem | Create or Overwrite File | **Preconditions:** None. **Steps:** Define content string and target `file_path` in Python → Use `with open(file_path, 'w') as f: f.write(content)` → Include error handling (`try-except`) to manage failures. **Notes:** Python file handling (`open`, `write`) is more robust than using shell commands (e.g., `echo`) due to escaping issues. |
| | Verify File Existence | **Preconditions:** None. **Steps:** (Python) Import `os` → iterate file paths → check with `os.path.exists()` → print per-file and summary results. (Alternative) Use `run_cmd` with `ls <file>` → verify from `returncode` and output. **Notes:** `os.path.exists` is suitable for programmatic checks; `ls` is effective for CLI verification with error messages. |
| OwnCloud | Login | **Preconditions:** Browser open, ownCloud URL & credentials available, service reachable. **Steps:** Go to login URL → handle connection errors (`ping`) if needed → refresh page → enter username/password via input fields → click `Log in` → verify login success (URL change, disappearance of fields, presence of post-login elements) → if modal appears, dismiss via `Escape`. **Notes:** Always pair navigation/input/click with `browser_update`. Use attributes like placeholder/text to locate elements. Verify success via URL and post-login UI, not just button clicks. |
| | Navigate to Folder by URL | **Preconditions:** Browser is open and authenticated. **Steps:** Navigate to folder URL → refresh state → verify URL and page title → dismiss modal (if any) via `Escape` → confirm presence of expected files in accessibility tree/interactive elements. **Notes:** Direct URL navigation is more reliable than clicking folder links. Always verify URL, title, and file list after navigation. |
| | | *... (additional entries omitted)* |

Table 8: Examples of $\mathcal{M}_{tool}$, providing optimized tool instructions and description.

| Tool | Description | Instruction |
|------|-------------|-------------|
| `access_guide` | Get detailed platform/application operation guides. The guide is structured from past successful experiences. - **Primary mode:** `batch_requests`, supporting single or multiple apps/items. - Example: `batch_requests={'RocketChat': ['Login', 'Create Channel']}`. - For single app/item: `application_name="RocketChat"` or with `item_names`. - Crucial: Always pass app name as dict key and items as list in `batch_requests`, otherwise `TypeError`. - Absence of requested guide entries is also a useful signal. - Guides may not match current UI exactly; adapt as needed. | Review the returned guide carefully. Compare actively with real-time UI observations. - If discrepancies appear, prioritize adaptive exploration. - Always check parameter names and types (`dict` for `batch_requests`, `str` for `application_name`) to avoid `TypeError`. |
| `browser_click` | Click interactive element on current browser page by index. - Always call `browser_update` first to ensure fresh indices. - Best practice: follow click with `browser_update` to confirm changes. - Prioritize semantic attributes (e.g., `text`, `aria-label`) over raw indices. - If clicking fails, dynamically search for element attributes. - Verify intended outcome (navigation, modal opening, state change), not just the click itself. - For persistent failures, consider `browser_send_keys('Enter')`. | Always follow with `browser_update`. - Verify outcome (e.g., page change, modal open). - If action fails, refresh interactive elements and retry with semantic attributes. - If still failing, try `browser_send_keys`. - For unclickable elements, acknowledge task may be unachievable and adjust strategy. |
| `browser_input` | Enter text into a specified browser element. - Always refresh elements first with `browser_wait_and_get_update`. - Input **appends** text; clear field with `""` if needed. - After input, changes may not persist without explicit `Save/Submit`. - Consider following input with `browser_update` to catch UI changes. | After input, always call `browser_update`. - Recheck interactive elements for changes. - If part of a form, explicitly locate and click `Save/Submit`. - Verify that the input was successfully saved. |
| *... (additional entries omitted)* | | |

## A.4   TOOL SET

We equip MUSE with a minimal yet sufficient tool set, consisting of a browser operator, a Python interpreter, a shell, a visual extractor, and a memory retriever. The browser operator is primarily implemented based on the browser-use framework Browser-Use (2025) , enhanced with both the accessibility (a11y) tree and the page's interactive elements as observations returned to the agent. The visual extractor leverages GPT-4o as the backbone model. A complete overview of the tool set is provided in Table 9.

Table 9: Tool Set.

| Tool | Function |
| --- | --- |
| run_cmd | Execute a full shell command string and return its result, suitable for file and system operations. |
| run_python_code | Execute Python code in an isolated environment for data processing and analysis. |
| access_guide | Retrieve structured procedural memory for accurate interaction. |
| gpt4o_describe_image | Use GPT-4o to recognize and interpret the content of images. |
| browser_go_to_url | Navigate the browser to a specified URL, supporting page refresh and reset. |
| browser_input | Input text into a specified field in the current browser page. |
| browser_send_keys | Send keyboard shortcuts or keystrokes (e.g., Enter) to the current browser tab. |
| browser_update | Wait and refresh to retrieve the latest accessibility tree and interactive elements. |
| browser_click | Click a specified interactive element in the current browser page by index. |
| browser_extract_content_by_vision | Extract specified content from a browser screenshot using GPT-4o. |
| browser_close_tab | Close a specified browser tab by index. |
| browser_go_back | Navigate back in the browser history of the current tab. |
| browser_list_tabs | List all currently open browser tabs. |
| browser_switch_tab | Switch to a specified browser tab by index. |

## A.5   THE GROWTH TREND OF TOKEN COST WITH SUBTASK EXECUTION.

In practice, we introduced several context management mechanisms, such as compressing redundant information like the Agent's early-generated code and the Browser Accessibility Tree. We conducted a statistical analysis of token consumption on the sampled set of 18 tasks $\mathcal{T}_{cl}$. Specifically, we calculated the cross-task mean of the average token overhead per action within subtasks. Furthermore, we also calculated the token count without context compression using the same method. Figure 8 and Figure 9 respectively show the token consumption during the training phases with or without context compression.

The results show that our context management mechanism can significantly reduce the number of tokens (achieving a compression rate of over ten-fold), effectively suppressing the explosive growth of context length and thus controlling the cost.

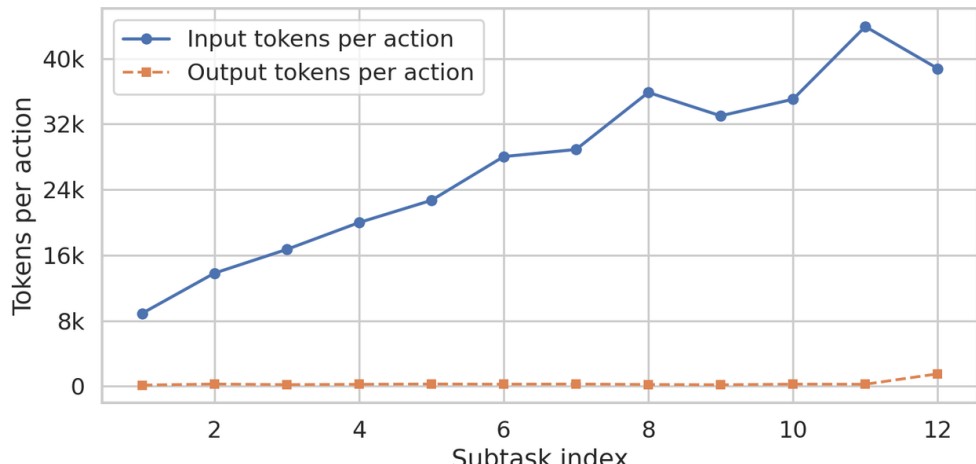

Figure 8: Token consumption statistics during training with context compression. We first calculated the average token consumption per action for each subtask within individual tasks. Subsequently, we computed the mean of this metric across 18 tasks for each corresponding subtask (i.e., the $k$-th subtask across all tasks).

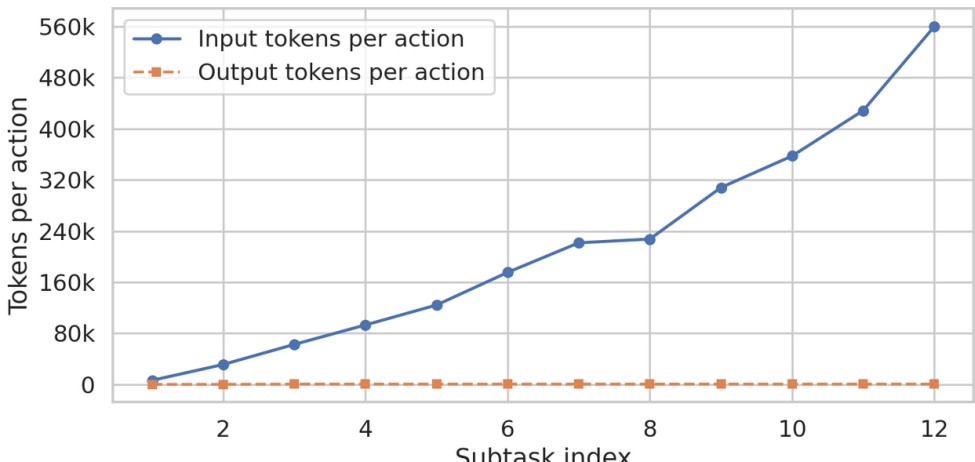

Figure 9: Token consumption statistics during training without context compression. We first calculated the average token consumption per action for each subtask within individual tasks. Subsequently, we computed the mean of this metric across 18 tasks for each corresponding subtask (i.e., the $k$-th subtask across all tasks).

## A.6 COMPLETE TASK-LEVEL RESULTS

Table 10 shows the scores of all TAC tasks. The overall scores and experimental analysis are shown in Section 4.3.3.

Table 10: Detailed results on the complete TAC benchmark for 175 tasks.

| Task | checkpoint | $S_{partial}$ (%) |
|------|------------|-------------------|
| admin-arrange-meeting-rooms | 0/2 | 0.0 |
| admin-ask-for-meeting-feedback | 6/6 | 100.0 |
| admin-ask-for-upgrade-reimbursement | 2/4 | 25.0 |
| admin-check-employees-budget-and-reply | 4/4 | 100.0 |
| admin-check-employees-budget-and-reply-2 | 4/4 | 100.0 |
| admin-check-employees-budget-and-reply-and-record | 6/6 | 100.0 |
| admin-collect-requests-and-compute-total-price | 1/4 | 12.5 |
| admin-employee-info-reconciliation | 5/7 | 35.71 |
| admin-get-best-vendor-quote | 5/6 | 41.67 |
| admin-make-spreadsheet | 0/5 | 0.0 |
| admin-mass-forms-filling | 0/5 | 0.0 |
| admin-read-survey-and-summarise | 2/3 | 33.33 |
| admin-remove-pages-pdf | 1/3 | 16.67 |
| admin-translate-sales-chat | 0/4 | 0.0 |
| admin-watch-video | 0/2 | 0.0 |
| bm-classify-nationality | 2/6 | 16.67 |
| ds-answer-numerical-data-question | 0/6 | 0.0 |
| ds-answer-spreadsheet-questions | 5/5 | 100.0 |
| ds-calculate-spreadsheet-stats | 2/5 | 20.0 |
| ds-coffee-shop-database-management | 4/10 | 20.0 |
| ds-find-meeting-spreadsheet | 1/2 | 25.0 |
| ds-fix-table-values-and-missing-answers | 6/6 | 100.0 |
| ds-format-excel-sheets | 3/4 | 37.5 |
| ds-janusgraph-exercise | 1/6 | 8.33 |
| ds-merge-multiple-sheets | 1/3 | 16.67 |
| ds-organise-report-sus-data | 3/5 | 30.0 |
| ds-predictive-modeling | 3/3 | 100.0 |
| ds-sql-exercise | 6/6 | 100.0 |
| ds-stock-analysis-slides | 1/8 | 6.25 |
| ds-visualize-data-in-pie-and-bar-chart | 4/4 | 100.0 |
| example | 3/5 | 30.0 |
| finance-apply-tax-credit | 0/8 | 0.0 |
| finance-budget-variance | 4/4 | 100.0 |
| finance-check-attendance-payroll | 3/3 | 100.0 |
| finance-create-10k-income-report | 1/6 | 8.33 |
| finance-expense-validation | 2/4 | 25.0 |
| finance-find-signatories | 2/5 | 20.0 |
| finance-invoice-matching | 1/5 | 10.0 |

| Task | checkpoint | $S_{partial}$ (%) |
|------|-----------|-------------------|
| finance-nonqualified-bill-ask-for-reimburse | 2/2 | 100.0 |
| finance-qualified-bill-ask-for-reimburse | 2/5 | 20.0 |
| finance-r-d-activities | 1/6 | 8.33 |
| finance-revenue-reconciliation | 1/4 | 12.5 |
| finance-substantial-presence-test | 1/2 | 25.0 |
| hr-analyze-outing-bills | 3/7 | 21.43 |
| hr-check-attendance-multiple-days | 1/4 | 12.5 |
| hr-check-attendance-multiple-days-department | 0/3 | 0.0 |
| hr-check-attendance-multiple-days-department-with-chat | 2/4 | 25.0 |
| hr-check-attendance-one-day | 3/3 | 100.0 |
| hr-check-for-invalid-passwords-and-ask-for-valid-passwords | 4/4 | 100.0 |
| hr-collect-feedbacks | 5/5 | 100.0 |
| hr-collect-multiple-valid-passwords | 2/4 | 25.0 |
| hr-create-career-ladder | 4/4 | 100.0 |
| hr-create-employee-manual | 1/4 | 12.5 |
| hr-delete-and-insert-user | 3/3 | 100.0 |
| hr-get-valid-password | 4/4 | 100.0 |
| hr-green-card-consultation | 3/3 | 100.0 |
| hr-internal-tooling-slides | 6/10 | 30.0 |
| hr-make-slides-introduce-leadership | 5/5 | 100.0 |
| hr-mass-survey | 1/7 | 7.14 |
| hr-massive-resume-screening | 5/5 | 100.0 |
| hr-new-grad-job-description | 3/3 | 100.0 |
| hr-new-grad-job-description-2 | 4/4 | 100.0 |
| hr-new-grad-job-description-3 | 5/5 | 100.0 |
| hr-organize-talent-info | 1/4 | 12.5 |
| hr-pick-interviewer-1 | 6/6 | 100.0 |
| hr-pick-interviewer-2 | 4/6 | 33.33 |
| hr-pick-interviewer-3 | 1/4 | 12.5 |
| hr-populate-salary-increase-memo | 4/7 | 28.57 |
| hr-resume-categorization | 1/4 | 12.5 |
| hr-resume-screening | 4/4 | 100.0 |
| hr-salary-analysis | 0/2 | 0.0 |
| hr-transfer-group | 1/3 | 16.67 |
| ml-generate-gradcam | 1/4 | 12.5 |
| ml-grade-exam | 1/8 | 6.25 |
| pm-add-new-moderator | 3/3 | 100.0 |
| pm-ask-for-issue-and-create-in-gitlab | 5/5 | 100.0 |
| pm-ask-issue-assignee-for-issue-status-and-update-in-plane | 3/3 | 100.0 |
| pm-assign-issues | 5/5 | 100.0 |
| pm-change-channel-ownership | 3/3 | 100.0 |
| pm-check-backlog-update-issues | 1/5 | 10.0 |
| pm-copy-plane-issues-to-gitlab | 3/4 | 37.5 |
| pm-create-channel-message | 3/3 | 100.0 |

| Task | checkpoint | $S_{partial}$ (%) |
|---|---|---|
| pm-create-channel-message-medium | 6/6 | 100.0 |
| pm-create-channel-new-leader | 2/3 | 33.33 |
| pm-create-plane-issue | 2/2 | 100.0 |
| pm-create-teammate-channel-from-spreadsheet | 4/5 | 40.0 |
| pm-distribute-information | 2/2 | 100.0 |
| pm-monitor-new-bug-issues | 2/4 | 25.0 |
| pm-monthly-attendance-slides | 4/4 | 100.0 |
| pm-plan-personnel-for-new-project | 3/7 | 21.43 |
| pm-prepare-meeting-with-customers | 6/6 | 100.0 |
| pm-present-engineer-group-members | 0/3 | 0.0 |
| pm-present-gitlab-info-as-ppt | 5/5 | 100.0 |
| pm-projects-analytics | 2/5 | 20.0 |
| pm-schedule-meeting-1 | 5/5 | 100.0 |
| pm-schedule-meeting-2 | 5/5 | 100.0 |
| pm-send-hello-message | 4/5 | 40.0 |
| pm-send-notification-to-corresponding-user | 4/4 | 100.0 |
| pm-update-gitlab-issue-from-plane-status | 2/3 | 33.33 |
| pm-update-plane-issue-from-gitlab-status | 7/7 | 100.0 |
| pm-update-project-milestones | 5/5 | 100.0 |
| pm-update-sprint-cycles | 3/4 | 37.5 |
| qa-escalate-emergency | 2/3 | 33.33 |
| qa-update-issue-status-according-to-colleagues | 6/6 | 100.0 |
| research-answer-questions-on-paper | 10/12 | 41.67 |
| research-reproduce-figures | 4/8 | 25.0 |
| sde-add-all-repos-to-docs | 4/7 | 28.57 |
| sde-add-one-gitlab-pipeline | 0/3 | 0.0 |
| sde-add-wiki-page | 4/4 | 100.0 |
| sde-change-branch-policy | 2/2 | 100.0 |
| sde-change-license-easy | 4/4 | 100.0 |
| sde-change-license-hard | 2/3 | 33.33 |
| sde-check-and-run-unit-test | 1/2 | 25.0 |
| sde-check-high-priority-issue | 1/4 | 12.5 |
| sde-close-all-gitlab-issues | 2/2 | 100.0 |
| sde-close-all-issue-on-all-project-under-tac-workspace | 2/3 | 33.33 |
| sde-close-all-prs | 2/2 | 100.0 |
| sde-close-an-issue | 2/2 | 100.0 |
| sde-collect-open-issues | 3/3 | 100.0 |
| sde-copilot-arena-server-easy-add-suffix | 4/4 | 100.0 |
| sde-copilot-arena-server-new-endpoint | 9/9 | 100.0 |
| sde-copilot-arena-server-setup | 7/7 | 100.0 |
| sde-copy-issues-to-plane | 2/2 | 100.0 |
| sde-copy-table-from-pdf-to-xlsx | 2/5 | 20.0 |
| sde-create-commit-table-for-all-gitlab-users | 1/6 | 8.33 |
| sde-create-new-characters | 2/4 | 25.0 |

| Task | checkpoint | $S_{partial}$ (%) |
|---|---|---|
| sde-create-new-gitlab-project-logo | 2/3 | 33.33 |
| sde-create-new-release | 2/2 | 100.0 |
| sde-create-new-repo | 2/3 | 33.33 |
| sde-create-sqlite-database | 6/8 | 37.5 |
| sde-debug-crashed-server | 2/8 | 12.5 |
| sde-delete-all-project-under-plane | 0/1 | 0.0 |
| sde-delete-all-repos | 1/1 | 100.0 |
| sde-delete-stale-branch | 2/2 | 100.0 |
| sde-dependency-change-1 | 5/5 | 100.0 |
| sde-find-answer-in-codebase-1 | 0/3 | 0.0 |
| sde-find-answer-in-codebase-2 | 3/3 | 100.0 |
| sde-find-answer-in-codebase-3 | 2/5 | 20.0 |
| sde-find-api | 2/4 | 25.0 |
| sde-fix-factual-mistake | 3/3 | 100.0 |
| sde-fix-rising-wave-datatype | 2/5 | 20.0 |
| sde-implement-buffer-pool-manager-bustub | 1/12 | 4.17 |
| sde-implement-covering-index-in-janusgraph | 0/3 | 0.0 |
| sde-implement-hyperloglog | 1/6 | 8.33 |
| sde-implement-raft-in-go | 0/10 | 0.0 |
| sde-install-go | 0/2 | 0.0 |
| sde-install-openjdk | 2/2 | 100.0 |
| sde-issue-label-management | 0/1 | 0.0 |
| sde-migrate-package-manager | 0/8 | 0.0 |
| sde-milestone-meeting | 2/5 | 20.0 |
| sde-move-bustub-wiki | 3/4 | 37.5 |
| sde-move-page-to-cloud | 2/3 | 33.33 |
| sde-pitch-idea-to-manager | 5/5 | 100.0 |
| sde-reply-community-issue-by-asking-npc | 5/5 | 100.0 |
| sde-reply-community-issue-with-fixed-reply | 3/3 | 100.0 |
| sde-repo_profile_pic | 1/3 | 16.67 |
| sde-report-agent-repos | 0/2 | 0.0 |
| sde-report-unit-test-coverage-to-plane | 3/4 | 37.5 |
| sde-run-all-unit-test | 3/4 | 37.5 |
| sde-run-janusgraph | 1/6 | 8.33 |
| sde-run-linter-on-openhands | 0/2 | 0.0 |
| sde-run-rising-wave-locally | 2/2 | 100.0 |
| sde-sotopia-create-agent | 5/5 | 100.0 |
| sde-sotopia-create-agent-wo-repo | 2/6 | 16.67 |
| sde-sotopia-dev-container | 2/7 | 14.29 |
| sde-sotopia-update-ci | 1/3 | 16.67 |
| sde-summarize-recent-issues | 4/4 | 100.0 |
| sde-sync-from-origin-repo | 1/1 | 100.0 |
| sde-troubleshoot-dev-setup | 1/4 | 12.5 |
| sde-update-dev-document | 4/4 | 100.0 |

| Task | checkpoint | $S_{partial}$ (%) |
|------|:----------:|:-----------------:|
| sde-update-issue-status-on-plane | 3/3 | 100.0 |
| sde-update-readme | 2/2 | 100.0 |
| sde-write-a-unit-test-for-append_file-function | 2/5 | 20.0 |
| sde-write-a-unit-test-for-scroll_down-function | 2/5 | 20.0 |
| sde-write-a-unit-test-for-search_file-function | 2/5 | 20.0 |

## A.7 USE OF LARGE LANGUAGE MODELS

Large language models were used solely as auxiliary tools for grammar check and linguistic polishing. All conceptual development, methodological design, experiments, and analysis were conducted entirely by the authors without reliance on LLMs.

