# OpenReview forum: "Learning on the Job: An Experience-Driven Self-Evolving Agent for Long-Horizon Tasks"
_ICLR.cc/2026/Conference — Submitted to ICLR 2026_

### Official Review · Reviewer_st2r · 2025-10-26

**Soundness:** 3
**Presentation:** 3
**Contribution:** 3
**Rating:** 6
**Confidence:** 3

**Summary:**

The authors propose MUSE, a novel self-evolving AI agent framework that enables LLMs to learn continuously from experience on real-world long-horizon tasks. It features a hierarchical memory module where the agent autonomously reflects on trajectories to create and integrate structured experience. This mechanism drives self-evolution, leading to new state-of-the-art performance on the TAC benchmark and demonstrating strong generalization to new tasks.

**Strengths:**

1) The experimental results demonstrate strong performance, achieving SOTA results compared to recent LLMs.
2) The work is well-motivated by the need for LLM agents to continuously learn from experience and overcome their static nature.

**Weaknesses:**

1) I wonder whether the performance gains are primarily derived from leveraging previous successful solutions. It is crucial to explore the framework's ability to learn from and generalize over failures. How about provide more harder questions in the initial and then easier questions, and will the model benifit from initial hard failure samples?
2) A primary concern for the framework is the computational overhead during inference. Can the authors provide inference time computation on the questions compared to without memory, and without MUSE?

**Questions:**

See weaknesses.

---

> ### Author Response · Authors · 2025-11-25
>
> Thank you very much for your detailed review and valuable feedback. We apologize for the slight delay in our response, which was necessary to incorporate additional experiments requested by other reviewers, including new model comparisons and evaluation on another new benchmark. If you are interested, please refer to our responses to other reviewers for the analysis and results of these experiments, such as our response to Reviewer aGFx's Q3A3.
>
> We thank you for your patience, and we hope our reply can resolve your concerns :)
>
> ---
>
> ## **Q1:** Sources of performance improvement.
>
> **A1:** We appreciate this keen observation. Indeed, the agent's performance gains primarily stem from the induction and utilization of past successful solutions. However, our framework also possesses the capability to learn from failure, although this currently constitutes a smaller portion of the learning process. This specific type of learning predominantly occurs in scenarios characterized by **"initial failure followed by eventual success"** (after reflection or retrying). In such cases, the agent records the critical information that led to the initial failure (i.e., the 'lessons') to optimize future decision-making.
>
> ## **Q2:** Can the model benefit from a setup that prioritizes difficulty over ease?
>
> **A2:** Regarding the hypothesis of prioritizing difficult tasks (hard-to-easy), this touches upon the core principle governing our task selection: **"Learnability."** We deliberately selected tasks of moderate difficulty to ensure that the agent retains the capacity to generate successful trajectories—via autonomous exploration—that serve as material for learning.
>
> If, as hypothesized, we were to provide only 'intractable samples' (which the agent is completely unable to solve) during the initial phase, the agent would be unable to accumulate any meaningful successful experiences to serve as an **"anchor"** for subsequent learning and generalization, particularly in the absence of **Human-in-the-loop** guidance. An experience library comprised exclusively of uncorrected failures would fundamentally impede the agent's evolution.
>
> In conclusion, we maintain that while learning from failure is crucial, a critical mass of **"solvable tasks"** (facilitating the accumulation of positive experience) is a necessary prerequisite for achieving autonomous evolution.
>
> ## **Q3:** Computational overhead during the reasoning process.
>
> **A3:** Following your suggestion, we conducted additional comparative experiments to evaluate the efficiency of different reasoning strategies. We compared the inference time across three configurations:
> -  MUSE framework with Accumulated Experience;
> -  MUSE framework without experience;
> -  Pure PE Agent (Baseline).
>
> The results are listed in the Table below. It is evident that although the reflection module and experience retrieval steps incur additional temporal overhead, they yield a substantial performance gain, justifying the trade-off.
>
> | Comparison of time for 18 tasks | Total time | Avg time | Sckpt (%) ↑ | Avg. Spartial (%) ↑ |
> | :--- | :--- | :--- | :--- | :--- |
> | test w/ memory | 17874.88s | 993.05s | 73.18 | 64.61 |
> | test w/o memory | 16113.38s | 895.19s | 66.12 | 55.85 |
> | test w/ pure PE Agent | 11161.26s | 620.07s | 63.53 | 43.21 |

---

> ### Author Response · Authors · 2025-11-28
> **Gentle Reminder: Feedback Request**
>
> We sincerely appreciate your valuable feedback on the paper. As the deadline for the discussion phase is rapidly approaching, we would be most grateful if you could **share any remaining concerns** at your earliest convenience, allowing us to address them immediately.

---

### Official Review · Reviewer_aGFx · 2025-10-27

**Soundness:** 3
**Presentation:** 4
**Contribution:** 3
**Rating:** 4
**Confidence:** 4

**Summary:**

MUSE proposes an agent framework with 3 modules : strategic, procedural, tool steps which refine knowledge in a predefined operation standard ( plan-execute-reflect-memorize ) scaffolding loop. When using MUSE on TheAgentCompany, it improves by over 20% from prior SOTA using Gemini-2.5-flash model.

**Strengths:**

MUSE is the first method to exceed 50% on TAC with a test time continuous learning module. The 3-tier structure (Strategic/Procedural/Tool) is intuitive and well-motivated, providing different levels of abstraction.

**Weaknesses:**

The paper doesn't adequately explain the details of the scaffolding specifically : How memory is deduplicated and pruned at scale? What happens when memories conflict? Computational cost of memory retrieval as size grows? ( These could be resolved with the inclusion of supplementary materials but its not provided )

The experiment setting is a bit weak with only one single benchmark ( the agent company ) and lacks the commonly used benchmarks such as AppWorld, OSWorld or even simpler task such as SWE-bench.

But more critically, the framework is not different enough from similar works which learns from trajectories ( ExpeL, Memp, Agent Workflow Memory), making the contributions too weak to justify its novelty.

**Questions:**

Major:

1. Would MUSE be applicable to SWE-bench, OSWorld, WebArena or AFLWorld as well?

2. Could you provide measurements of latency, token costs, or context growth over time? Given memory updates after every sub-task, computational overhead could be prohibitive

3. How sensitive is performance to the quality of the Reflect Agent? What if it extracts poor memories ( by using smaller models on reflect agent)?

4. What’re the error failure case where MUSE still fails to perform successfully? For example in the original TAC benchmark, the author identifies lack of social skills, browsing, self-confusion are one of the 3 major causes of error.

Minor:

5. How long does it take to craft the prompts for each of the modules? Does the time needed to craft each of the module prompts to adapt to new LLMs or tasks hinders the adoption of MUSE?

6. Is there a solid reason why the supplementary material is not provided? Without the code it is impossible to review if the evaluation does not contain any information leak from ground truth during the iterative improvement process.

7. In the paper, it claims MUSE is model-agnostic memory, but were the memories actually tested when transferred to significantly different model families ( deepseek-3.1 -> gemini 2.5 flash )?

---

> ### Author Response · Authors · 2025-11-25
> **Response to Reviewer aGFx (1/4)**
>
> Thank you very much for your detailed review and valuable feedback. We have incorporated your suggestions, particularly by including additional experiments on another benchmark, and we hope these address your concerns.
>
> This response was a little delayed due to the substantial time required for the additional experiments. Thank you for your understanding :)
>
> ---
>
> ## **Q1:** Question about memory maintenance and retrieval efficiency.
>
> **A1:** The agent autonomously maintains its entire memory repository, performing operations such as deletion, modification, and merging. In our TAC experiments, we have not observed large-scale data redundancy; thus, no specialized de-duplication handling has been necessary so far.
>
> Memory conflicts are categorized into the following scenarios:
>
> - **Conflicts involving erroneous memory:** If the agent utilizes a memory entry that leads to task failure, it will evaluate the entry and opt to delete it.
> - **Conflicts between valid memories with differing preconditions:** In cases where conflicting memories are both correct but apply to different contexts, the agent explicitly lists their respective preconditions. This annotation assists the agent in making informed judgments and selections in future tasks.
>
> **Regarding memory retrieval**, we employ efficient context management mechanisms to mitigate computational costs. Specifically, once the agent selects relevant experience entries based on the directory index, redundant information (such as the directory itself) is discarded from the context. This ensures that the retrieval process does not incur significant overhead.
> We have included an additional **Table** to compare the inference time and performance between the agent with and without experience utilization. The memory bank used in this experiment comprises 10 Strategy Memory, 14 Tool Memory, and 75 Procedure Memory. As shown in the results, after a period of experience accumulation, incorporating memory results in only a **marginal increase in average inference time (from 895.19s to 993.05s)**. However, this slight trade-off yields a **substantial performance improvement (score increasing from 55.85 to 64.61)**, demonstrating the high efficiency of our approach.
>
> | Comparison of time for 18 tasks | Total time | Avg time | Sckpt (%) ↑ | Avg. Spartial (%) ↑ |
> | :--- | :--- | :--- | :--- | :--- |
> | test w/ memory | 17874.88s | 993.05s | 73.18 | 64.61 |
> | test w/o memory | 16113.38s | 895.19s | 66.12 | 55.85 |
>
> ## **Q2:** The growth trend of token cost with subtask execution.
>
> **A2:** In practice, we introduced several **context management mechanisms**, such as compressing redundant information like the Agent's early-generated code and the Browser Accessibility Tree.
>
> We conducted a statistical analysis of token consumption on the sampled set of 18 tasks $\mathcal{T}_{cl}$. Specifically, we calculated the cross-task mean of the average token overhead per action within subtasks. Furthermore, we also calculated the token count without context compression using the same method. Detailed visualizations are provided in **Appendix A.5**.
>
> The results show that our context management mechanism can **significantly reduce the number of tokens (achieving a compression rate of over ten-fold)**, effectively suppressing the explosive growth of context length and thus controlling the cost.

---

> ### Author Response · Authors · 2025-11-25
> **Response to Reviewer aGFx (2/4)**
>
> ## **Q3:** Select a new benchmark and compare it with other methods.
>
> **A3:**   We thank you for your insightful comments. Establishing a direct lateral comparison was challenging due to the absence of other self-evolving agents already tested on the TAC benchmark and the prohibitively high engineering costs required to adapt existing open-source methods.
> However, to address your concerns, we conducted additional experiments on a widely adopted benchmark, **WebArena**, specifically comparing our approach with the *Agent Workflow Memory (AWM)* method.
>
> Compared to TAC, WebArena benchmark focuses more on single-platform, short-cycle tasks, which to some extent limits the full realization of the MUSE framework.
> Due to the tight timeframe of the rebuttal, we performed a **minimal migration**: the entire code adaptation process took only about 15 minutes, limited to adjusting the task background in the prompt and adapting basic WebArena tools (web actions, code interface, and memory retrieval). Crucially, other methods, including AWM, used the **superior browser operations and observation tool BrowserGym**, while we only used the tools natively provided by WebArena. Moreover, we **did not integrate visual recognition tools in this experiment**, which objectively rendered certain vision-dependent tasks impossible to complete. In summary, no benchmark-specific parameter optimization or deep adaptation was applied for WebArena. The final results are presented in the table below:
>
> |            Method           | Success Rate (%) |
> |:---------------------------:|:----------------:|
> |      Claude Code + MCP      |        68        |
> |       OpenAI Operator       |       58.1       |
> |        AgentSymbiotic       |       52.1       |
> |           WebPilot          |       37.2       |
> |     GUI-API Hybrid Agent    |       35.8       |
> |    AWM + GPT-4-0613    |       35.5       |
> |      BrowserGym + GPT-4     |       23.5       |
> |      GPT-4 + Auto Eval      |       20.2       |
> |     GPT-4o + Tree Search    |       19.2       |
> |          gpt-4-0613         |       14.9       |
> | **Muse + gemini-2.5 flash** |     **38.9** |
>
>
> As shown in the table, using MUSE with Gemini-2.5 Flash, we performed memory evolution over just two iterations on 80 tasks (without any Ground Truth supervision) and subsequently achieved excellent results across all 812 test tasks. Most critically, despite the lack of specific adaptation, **MUSE outperformed the comparison method AWM + GPT-4-0613**. (Note: Considering the currently high cost of GPT-4, this experiment maintained the use of the more cost-effective Gemini-2.5 Flash, consistent with the main paper).
>
> Compared to AWM, MUSE incorporates a more advanced **hierarchical heterogeneous memory mechanism**. MUSE's methodology aligns more closely with human cognitive models; it forms a closed-loop, self-evolving complex system by learning not only "how to act" (  $M_{proc}$ )  but also "how to think and plan" ($M_{strat}$) and "how to optimize tool usage" (${M}_{tool}$). This design has demonstrated superior performance on both WebArena and the more complex, longer-horizon TAC benchmark, proving the importance of MUSE's efficient memory mechanism and verifying its immense potential for continual learning in realistic, long-term tasks.
>
> MUSE is fundamentally different from other methods in the table that rely on business 'computer use' LLMs (e.g., OpenAI Operator and Claude Code) or require massive fine-tuning data (e.g., AgentSymbiotic).
> The core mechanism of MUSE lies in distilling interaction trajectories into generalizable procedural knowledge and strategies to prevent repeated exploration and error accumulation in complex, long-cycle tasks.

---

> ### Author Response · Authors · 2025-11-25
> **Response to Reviewer aGFx (3/4)**
>
> ## **Q4:** The impact of the Reflect Agent on performance.
>
> **A4:** First, **Table 3** in the manuscript has already demonstrated that removing the Reflect Agent leads to a degradation in performance, as it bears the critical responsibility of filtering errors and distilling SOPs.
>
> Second, to further investigate the impact of the Reflect Agent and the risks associated with "maladaptive memories," we conducted additional experiments using the smaller **Gemini-2.5 Flash Lite** model in the **Table below**. These experiments revealed a critical phenomenon we term the **"Capability Threshold"**:
>
> - When employing a smaller model (Gemini-2.5 Flash Lite) as the Reflect Agent, performance showed a **transient improvement** in Iteration 2 (55.71%) but subsequently regressed in Iteration 3 (42.28%).
>
> - This confirms that if the model's capability falls below the necessary threshold, its reflective mechanism is insufficient to filter noise. Consequently, the accumulation of erroneous memories leads to a performance collapse.
>
> We posit that **effective self-evolution requires the backbone model to possess a foundational level of reasoning capability** to sustain a virtuous cycle. For models with weaker capabilities, the natural language memory format supported by MUSE facilitates the introduction of human feedback for **"memory cleaning"** (or curation), serving as an effective supplementary mechanism to overcome this bottleneck.
>
> | | checkpoint | Sckpt (%) ↑ | Avg. Spartial (%) ↑ | Perfect Completion Rate(%) ↑ |
> | :--- | :---: | :---: | :---: | :---: |
> | iter1 | 50/85 | 58.82 | 46.18 | 33.33 |
> | iter2 | 54/85 | 63.53 | 55.71 | 44.44 |
> | iter3 | 46/85 | 54.12 | 42.28 | 27.78 |
>
> ## **Q5:** Failure cases analysis.
>
> **A5:** Thanks for your insightful comment, you raised a good question.
>
> First, **certain complex Ultra-Long-Horizon Tasks impose exacting demands on the agent's long-term memory and reasoning capabilities**, thereby limiting the overall completion rate. Notably, to compel MUSE to transform successful solutions into reusable Procedural Memory, we intentionally provided **only a minimal toolset** in our experiments. While this constraint effectively showcased the agent's learning capabilities, it objectively increased the complexity and burden of the agent's web interactions.
>
> Second, **limitations in environmental observation constitute a critical factor**. For instance, a lack of **anticipatory perception** regarding unstructured data (e.g., images embedded in PDFs) causes processing pipeline failures. Similarly, insufficient observation of structured data formats (e.g., thousand separators in Excel) results in the inability to extract valid numerical values during the code generation and execution phases.
>
> **These failure scenarios highlight a fundamental disparity in perceptual modes between agents and humans**: Humans possess **proactive intuition** and **parallel multimodal perception**, enabling the instantaneous construction of global cognition. In contrast, agent perception remains **passive and linear**; lacking prior environmental intuition, agents frequently encounter blind spots when navigating non-standard data or complex environments.
>
> ## **Q6:** Prompt Development Time and Prompt Transferability.
>
> **A6:** We devoted approximately four days to iterative experimentation and testing to finalize the prompts for all modules. Our design was centered on **maximally decoupling** benchmark-specific characteristics. This strategy significantly reduced the effort required for **benchmark transfer**: typically, it requires adjusting the task background and related profile in the system prompts, as well as the callable toolset corresponding to the benchmark, while the remainder allows for direct reuse.
>
> Notably, in the TAC model transfer experiments detailed in **Section 4.4.2**, the system maintained robust performance even when switching from Gemini to DeepSeek using **unmodified prompts**. Naturally, to achieve optimal efficacy, tailoring prompts to the specific target LLM (e.g., optimizing Function Call formats) would help further unlock the model's performance potential.

---

> ### Author Response · Authors · 2025-11-25
> **Response to Reviewer aGFx (4/4)**
>
> ## **Q7:** Does the iterative improvement process include ground truth information?
>
> **A7:** We have provided two screen-recorded demonstration videos in Supplementary Materials. Regrettably, due to time constraints on submission, we were unable to fully prepare and submit the associated codebase concurrently with this manuscript. **We have now updated the supplementary material and included the complete source code.**
>
> We explicitly guarantee that throughout the entire continuous learning process, the experimental model was **strictly isolated** from Ground Truth information of all tasks, ensuring that **no form of data leakage occurred**. The GT data was employed **solely** during the final performance evaluation phase.
>
> ## **Q8:** The ability to transfer accumulated experience between different models
>
> **A8:** We apologize for an oversight in **Section 4.4.2**, where we did not sufficiently clarify a critical experimental detail:
>
> In the "DeepSeek-V3 w/ memory" experiment, the memory utilized was precisely the experience library accumulated by MUSE + Gemini-2.5 Flash after three rounds of continuous learning on $\mathcal{T}_{cl}$.
>
> This experimental setup provides compelling evidence for our assertion that the experience accumulated by MUSE is both **model-agnostic and generalizable**. We will explicitly clarify this distinction in the revised manuscript.

---

> ### Comment · Reviewer_aGFx · 2025-11-27
>
> ## Q3 new evaluation
>
> I appreciate the additional experiment, but could gemini-2.5-flash be directly comparable with other baselines?  As shown on your leaderboard, the majority of other methods use either gpt-4o as their backbone ( WebPilot, OpenAI Operator ) or claude 3.5 ( AgentSymbiotic ). Unfortunately, I do not have enough information to asses whether this leaderboard is valid or not, only that MUSE with gemini 2.5 flash could run on WebArena.
>
> ### Q4 : impact of agent
>
> > Table 3 in the manuscript has already demonstrated that removing...
>
> I'm aware of the result, but what I wish to see if the agent is replaced with a weaker one like gemini-2.5-flash-lite.
>
> > We posit that effective self-evolution requires the backbone model to possess a foundational level of reasoning capability..
>
> But this is a new finding right? Meaning the reflection agent should be as powerful as possible?
>
>
> ## Q5: Failure cases analysis.
>
> Do you have a more detailed case study? Like a test sample from TAC or WebArena.
>
>
> ## Other questions
>
> I'm satisfied with the answer, but these are minor concerns.

---

> > ### Author Response · Authors · 2025-11-27
> >
> > We sincerely appreciate your valuable feedback. We are pleased that the majority of your questions have been addressed to your satisfaction.
> >
> > ## Response to Q3 new evaluation
> >
> > In selecting an alternative benchmark, we focused on **three key criteria**: popularity, feasibility for evaluating continuous learning, and the availability of other continuous learning agents for direct comparison.
> > This selection process severely narrowed our options, ultimately leading us to the WebArena benchmark.
> >
> > However, official test results for Gemini-2.5 Flash on this platform are currently unavailable. Given that the WebArena benchmark comprises over 800 tasks, we found it infeasible to conduct additional testing experiments within the limited rebuttal window.
> >
> > Based on our observation that the pure GPT-4 model achieved a score of only around 20, we conservatively estimate that the pure Gemini-2.5 Flash model would likely not exceed 30 points.
> >
> > We deeply appreciate your understanding of the experimental limitations during this tight rebuttal period.
> >
> > ## Response to Q4: Impact of Agent and Self-Evolution
> >
> > A4: Thank you for this follow-up. You are correct, and we appreciate you highlighting this implication.
> >
> > 1. Confirmation of the "Capability Threshold."
> >
> > The experiment results we provided in our previous response `(Response to Reviewer aGFx 3/4)` exactly follow your instructions, where we replaced the Reflect Agent with a weaker **Gemini-2.5-Flash-Lite** model. The performance regression observed in later iterations actually validated one of our hypotheses (or maybe a new finding): the Reflect Agent must possess a foundational level of reasoning capability. Otherwise, the "self-evolution" mechanism may introduce errors rather than correct them.
> >
> > 2. Impact of a Stronger Backbone (Gemini-2.5 Pro)
> >
> > Conversely, replacing the backbone with a stronger model significantly enhances the quality of reflection. As detailed in our `Response to Reviewer jjy4 (2/2)`, when we conducted additional experiments and upgraded both the PE Agent and Reflect Agent to **Gemini-2.5 Pro**, the performance improved dramatically, achieving an 80%  S_ckpt.
> >
> > | Framework | Model | Checkpoint | S_ckpt (%) ↑ | Avg. S_partial (%) ↑ |
> > | :--- | :--- | :--- | :--- | :--- |
> > | OpenHands | gemini-2.5 pro | 58 / 85 | 65.39 | 57.67 |
> > | MUSE | gemini-2.5 flash | 62.2 / 85 | 73.18 | 64.61 |
> > | **MUSE** | **gemini-2.5 pro** | **68 / 85** | **80.00** | **67.10** |
> >
> > 3. Design Philosophy: Self-Evolution vs. Model Distillation
> >
> > Regarding your specific question—*"Meaning the reflection agent should be as powerful as possible?"*—while using a stronger model for reflection undoubtedly yields better performance (as shown with Pro), our core design philosophy for MUSE differs slightly.
> >
> > We specifically used the **same model** for both the PE Agent and the Reflect Agent to **avoid** making MUSE behave like a **"distillation framework"**. (i.e., where a weaker "student" PE Agent is guided by a significantly stronger "teacher" Reflect Agent). Instead, our goal was to demonstrate true **Self-Evolution**: verifying that an agent can break through its own capability limits using its own reasoning faculties. We believe this setup provides a more rigorous proof of the agent's ability to learn and evolve autonomously.
> >
> > ## Response to Q5: Failure cases analysis.
> >
> > We will present the two failure cases referenced in A5, but it will take some time to compile them into a clear and easily digestible format them into a clear and easily digestible format.
> >
> > We are prioritizing this task and will share the results with you immediately upon completion：)
> >
> > We hope this context helps clarify your concerns.

---

> ### Author Response · Authors · 2025-11-28
>
> ## Response to Q5: Failure cases analysis.
>
> We have incorporated the two specific TAC failure cases into the latest paper version. These detailed examples can be found in Figures 6 and 7 of the Appendix. (page 19 & 20)
>
> We hope these examples offer a clearer illustration of the failure scenarios.

---

### Official Review · Reviewer_jjy4 · 2025-10-30

**Soundness:** 2
**Presentation:** 3
**Contribution:** 2
**Rating:** 4
**Confidence:** 2

**Summary:**

This work creates an framework that allows a model to learn from experience for long-horizon tasks using a memory module. They achieve state-of-the-art performance on the TAC benchmark.

**Strengths:**

- Addresses timely concern on models being static and not learning as they perform the task
- Achieves new SOTA on TAC benchmark
- Demonstrates that memory module is memory-agnostic and can then be plugged into different models

**Weaknesses:**

- Only use of one domain; while i understand that they need a difficult domain with a long horizon, it would be important to see what how this agent would perform on other problems
- The questionable choice of baselines, they all use different base models and none of the works in the related works were experimented with
- It is important to discuss the time completing tasks and the time it takes to train

**Questions:**

Figure 3 is hard to read

---

> ### Author Response · Authors · 2025-11-25
> **Response to Reviewer jjy4 (1/2)**
>
> Thank you very much for your detailed review and valuable feedback. We have incorporated your suggestions, particularly by including additional experiments on another benchmark, and we hope these address your concerns.
>
> This response was a little delayed due to the substantial time required for the additional experiments. Thank you for your understanding :)
>
> ---
>
> ## **Q1:** Additional experiments on another benchmark.
>
> **A1:** Establishing a direct lateral comparison was challenging due to the absence of other self-evolving agents tested on TAC and the prohibitively high engineering costs to adapt existing open-source methods.
> However, to address your concerns, we conducted additional experiments on a widely adopted benchmark, **WebArena**, specifically comparing our approach with the *Agent Workflow Memory (AWM)*.
>
> Compared to TAC, WebArena focuses more on single-platform, short-cycle tasks, which to some extent limits the full realization of the MUSE framework.
> Due to the tight timeframe of the rebuttal, we performed a **minimal migration**: the entire code adaptation took only about 15 minutes, limited to adjusting the task background in the prompt and adapting basic WebArena tools (web actions, code interface, and memory retrieval). Crucially, while other methods, including AWM, used the **superior browser operations and observation tool BrowserGym**, we only used the native tools provided by WebArena. Moreover, we did not integrate visual recognition tools, which objectively rendered certain vision-dependent tasks impossible to complete. In summary, no benchmark-specific parameter optimization or deep adaptation was applied for WebArena. The final results are presented below:
>
> |            Method           | Success Rate (%) |
> |:---------------------------:|:----------------:|
> |      Claude Code + MCP      |        68        |
> |       OpenAI Operator       |       58.1       |
> |        AgentSymbiotic       |       52.1       |
> |           WebPilot          |       37.2       |
> |     GUI-API Hybrid Agent    |       35.8       |
> |    AWM + GPT-4-0613    |       35.5       |
> |      BrowserGym + GPT-4     |       23.5       |
> |      GPT-4 + Auto Eval      |       20.2       |
> |     GPT-4o + Tree Search    |       19.2       |
> |          gpt-4-0613         |       14.9       |
> | **Muse + gemini-2.5 flash** |     **38.9** |
>
>
> As shown in the table, using MUSE with Gemini-2.5 Flash, we performed memory evolution over just 2 iterations on 80 tasks (without any GT supervision) and subsequently achieved excellent results across all 812 test tasks. Most critically, despite the lack of specific adaptation, **MUSE outperformed the comparison method AWM + GPT-4-0613**. (Note: Considering the currently high cost of GPT-4, this experiment maintained the use of the more cost-effective Gemini-2.5 Flash, consistent with the main paper).
>
> Compared to AWM, MUSE incorporates a more advanced **hierarchical heterogeneous memory mechanism**. MUSE's methodology aligns more closely with human cognitive models; it forms a closed-loop, self-evolving complex system by learning not only "how to act" ($M_{proc}$)  but also "how to think and plan" ($M_{strat}$) and "how to optimize tool usage" (${M}_{tool}$). This design has demonstrated superior performance on both WebArena and the more complex, longer-horizon TAC benchmark, proving the importance of MUSE's efficient memory mechanism and verifying its immense potential for continual learning in realistic, long-term tasks.
>
> MUSE is fundamentally different from other methods in the table that rely on business 'computer use' LLMs (e.g., OpenAI Operator and Claude Code) or require massive fine-tuning data (e.g., AgentSymbiotic).
> The core mechanism of MUSE lies in distilling interaction trajectories into generalizable procedural knowledge and strategies to prevent repeated exploration and error accumulation in complex, long-cycle tasks.
>
> ## **Q2:** Time required to complete the task and for training.
>
> **A2:** We quantified the temporal costs of the training phase (experience accumulation) versus the testing phase (inference utilizing experience) across 18 tasks, as detailed in the Table below. Furthermore, we compared the time and performance of using experience-based reasoning versus not using experience-based reasoning. While the training phase inevitably introduces additional time overhead, our comparative analysis of inference—with and without experience—reveals a crucial insight: once the experience library is established, retrieving and utilizing these insights incurs **negligible additional latency** during inference, yet results in a **substantial enhancement** in agent performance.
>
> | Comparison of time for 18 tasks | Total time | Avg time | Sckpt (%) ↑ | Avg. Spartial (%) ↑ |
> | :--- | :--- | :--- | :--- | :--- |
> | train | 26419.86s | 1467.77s | -- | -- |
> | test w/ memory | 17874.88s | 993.05s | 73.18 | 64.61 |
> | test w/o memory | 16113.38s | 895.19s | 66.12 | 55.85 |

---

> ### Author Response · Authors · 2025-11-25
> **Response to Reviewer jjy4 (2/2)**
>
> ## **Q3:** Different models were used in the baselines.
>
> **A3:** Due to the prohibitive time and cost associated with evaluating the full TAC benchmark, the comparative data presented in our tables are derived from official TAC evaluation results. Notably, despite **using the Gemini-2.5 Flash model, our approach outperformed the intrinsically stronger Gemini-2.5 Pro model implemented with the OpenHands framework**. This result strongly underscores the superiority of our architecture.
>
> Furthermore, based on your suggestion, we conducted an **additional experiment** by replacing the base model in the MUSE framework with the more capable **Gemini-2.5 Pro** (using the exact same accumulated experience as the Gemini-2.5 Flash experiment). We tested this configuration on the same 18-task subset $\mathcal{T}_{cl}$ .  The results, shown in the Table, indicate a remarkable performance improvement when the model is upgraded to the stronger Gemini-2.5 Pro under identical experimental conditions.
>
> | Framework | Model | checkpoint | Sckpt (%) ↑ | Avg. Spartial (%) ↑ |
> | :---: | :---: | :---: | :---: | :---: |
> | OpenHands | gemini-2.5 pro | 58 / 85 | 65.39 | 57.67 |
> | MUSE | gemini-2.5 flash | 62.2 / 85 | 73.18 | 64.61 |
> | MUSE | gemini-2.5 pro | **68 / 85** | **80** | **67.1** |
>
> ## **Q4:** Figure 3 is hard to read.
>
> **A4:** The horizontal dashed line in Figure 3 represents the performance without using memory, serving as a baseline for comparison. The solid broken line represents the task performance statistically analyzed after each round of training (accumulating experience while performing tasks), showing a continuous upward trend, indicating that the Muse framework has continuous learning capabilities. We hope this explanation offers a clearer understanding of the image.

---

> ### Author Response · Authors · 2025-11-28
> **Gentle Reminder: Feedback Request**
>
> We sincerely appreciate your valuable feedback on the paper. As the deadline for the discussion phase is rapidly approaching, we would be most grateful if you could **share any remaining concerns** at your earliest convenience, allowing us to address them immediately.

---

### Official Review · Reviewer_SE9S · 2025-10-31

**Soundness:** 2
**Presentation:** 2
**Contribution:** 2
**Rating:** 2
**Confidence:** 5

**Summary:**

This paper proposes MUSE, an experience-driven agent framework with a hierarchical memory module for self-evolution in long-horizon tasks.

**Strengths:**

MUSE achieves a new SOTA on the TAC benchmark using a lightweight model, demonstrating effective continuous learning and generalization.

**Weaknesses:**

1. The limitations of existing methods or the motivation of this paper are not entirely accurate. The limitations of existing methods mentioned in the abstract—"they are test-time static and cannot learn from experience, lacking the ability to accumulate knowledge and continuously improve on the job"—can be addressed by an RL-based LLM.
2. The experimental comparisons are incomplete and unfair.

**Questions:**

1. The limitations of existing methods mentioned in the abstract—"they are test-time static and cannot learn from experience, lacking the ability to accumulate knowledge and continuously improve on the job"—can be addressed by LLM based on RL. However, the limitations of existing methods or the motivation in this paper are not entirely accurate.
2. For experience-driven self-evolving agents, how large is the amount of accumulated experience data? What is the scale of structured experience? How can we ensure that this experience is not forgotten during task switching?
3. Many self-evolving agents have not been compared. You can search for surveys of Self-Evolving Agents.
4. Why was only a subset of 18 tasks used for continuous learning experiments, and how does this selection bias affect the claim of general self-evolution capability?
5. How does the hierarchical memory structure ensure that retrieved experiences are contextually relevant and do not introduce noise or outdated strategies?
6. Can you justify the fairness of comparing MUSE using Gemini-2.5 Flash against stronger models in baseline methods without ablation on model capacity?

---

> ### Author Response · Authors · 2025-11-25
> **Response to Reviewer SE9S (1/3)**
>
> Thank you very much for your detailed review and valuable feedback. We have incorporated your suggestions, particularly by including additional experiments on another benchmark, and we hope these address your concerns.
>
> This response was a little delayed due to the substantial time required for the additional experiments. Thank you for your understanding :)
>
> ---
>
> ## **Q1:** Using RL to solve the test-time static issue.
>
> **A1:** We thank the reviewer for the insightful feedback. We fully agree that Reinforcement Learning (RL) is a critical method for enhancing model capabilities. However, the limitations we address in existing methods, specifically under the constraints of **Long-Horizon Productivity Tasks**, focus on two pain points that RL-based approaches currently find challenging to solve directly:
>
> 1. **Test-time Evolution vs. Training-time Optimization**: RL methods typically rely on parameter updates (fine-tuning). Once an RL-optimized model is deployed, its policy is difficult to update dynamically at runtime, making it primarily a Training-time Optimization approach. In contrast, our MUSE achieves continuous improvement by updating an **explicit memory module** instead of the model's parameters. This mechanism enables the Agent to adapt instantly to dynamic environments at **test-time**, providing a clear advantage.
> 2. Sample Efficiency and Black-Box Constraints: Long-horizon tasks require dozens, or even over a hundred, actions to complete. This scenario leads to extremely sparse rewards and high trial-and-error costs for RL-based training. Furthermore, RL fine-tuning cannot be easily applied to powerful closed-source models (like Gemini or ChatGPT). In comparison, MUSE's autonomous reflection mechanism (**Reflect Agent**) can consolidate successful or failed trajectories into reusable Standard Operating Procedures with very few attempts. This process requires no external reward signal or human intervention, demonstrating significantly higher sample efficiency.
>
> In the revised manuscript, we will more rigorously define the "limitations of existing methods," clearly distinguishing between RL optimization during the training phase and the low-resource, high-efficiency "on-the-job" learning scenarios at test time that MUSE focuses on.
>
> ## **Q2:** Regarding the amount of experience and how to ensure that experience is not forgotten.
>
> **A2:** In the TAC experiments, the size of our structured experience is detailed as follows:
> - The **Strategy Memory** is limited to fewer than 10 entries (this is controlled by a specific hyperparameter setting).
> - The **Procedure Memory** accumulates approximately 100 entries.
> - The **Tool Memory** size is fixed at 14, which corresponds to the total number of available tools.
>
> The three types of experience are stored separately in **text format** and are loaded into the agent's context using distinct mechanisms (detailed specifications on the loading protocols for each memory type are provided in Section 3.2). This approach ensures that experience remains persistent and is neither lost nor forgotten.

---

> ### Author Response · Authors · 2025-11-25
> **Response to Reviewer SE9S (2/3)**
>
> ## **Q3:** Select a new benchmark and compare it with other methods.
>
> **A3:** Thank you for the insightful feedback. Establishing a direct lateral comparison was challenging due to the absence of other self-evolving agents tested on TAC and the prohibitively high engineering costs to adapt existing open-source methods.
> However, to address your concerns, we conducted additional experiments on a widely adopted benchmark, **WebArena**, specifically comparing our approach with the *Agent Workflow Memory (AWM)*.
>
> Compared to TAC, WebArena focuses more on single-platform, short-cycle tasks, which to some extent limits the full realization of the MUSE framework.
> Due to the tight timeframe of the rebuttal, we performed a **minimal migration**: the entire code adaptation took only about 15 minutes, limited to adjusting the task background in the prompt and adapting basic WebArena tools (web actions, code interface, and memory retrieval). Crucially, while other methods, including AWM, used the **superior browser operations and observation tool BrowserGym**, we only used the native tools provided by WebArena. Moreover, we did not integrate visual recognition tools, which objectively rendered certain vision-dependent tasks impossible to complete. In summary, no benchmark-specific parameter optimization or deep adaptation was applied for WebArena. The final results are presented below:
>
> |            Method           | Success Rate (%) |
> |:---------------------------:|:----------------:|
> |      Claude Code + MCP      |        68        |
> |       OpenAI Operator       |       58.1       |
> |        AgentSymbiotic       |       52.1       |
> |           WebPilot          |       37.2       |
> |     GUI-API Hybrid Agent    |       35.8       |
> |    AWM + GPT-4-0613    |       35.5       |
> |      BrowserGym + GPT-4     |       23.5       |
> |      GPT-4 + Auto Eval      |       20.2       |
> |     GPT-4o + Tree Search    |       19.2       |
> |          gpt-4-0613         |       14.9       |
> | **Muse + gemini-2.5 flash** |     **38.9** |
>
>
> As shown in the table, using MUSE with Gemini-2.5 Flash, we performed memory evolution over just 2 iterations on 80 tasks (without any GT supervision) and subsequently achieved excellent results across all 812 test tasks. Most critically, despite the lack of specific adaptation, **MUSE outperformed the comparison method AWM + GPT-4-0613**. (Note: Considering the currently high cost of GPT-4, this experiment maintained the use of the more cost-effective Gemini-2.5 Flash, consistent with the main paper).
>
> Compared to AWM, MUSE incorporates a more advanced **hierarchical heterogeneous memory mechanism**. MUSE's methodology aligns more closely with human cognitive models; it forms a closed-loop, self-evolving complex system by learning not only "how to act" (  $M_{proc}$ )  but also "how to think and plan" ($M_{strat}$) and "how to optimize tool usage" (${M}_{tool}$). This design has demonstrated superior performance on both WebArena and the more complex, longer-horizon TAC benchmark, proving the importance of MUSE's efficient memory mechanism and verifying its immense potential for continual learning in realistic, long-term tasks.
>
> MUSE is fundamentally different from other methods in the table that rely on business 'computer use' LLMs (e.g., OpenAI Operator and Claude Code) or require massive fine-tuning data (e.g., AgentSymbiotic).
> The core mechanism of MUSE lies in distilling interaction trajectories into generalizable procedural knowledge and strategies to prevent repeated exploration and error accumulation in complex, long-cycle tasks.
>
> ## **Q4:** Discuss the use of 18 tasks for continuous learning and the resulting bias on the claim of general self-evolution.
>
> **A4:** Thank you for your question. The selection of 18 tasks was primarily driven by cost considerations. Running continuous learning experiments on the entire task set would be unacceptable in terms of both time and cost. Furthermore, accumulating experience on a subset of tasks allowed us to better test the generalization capability of the learned memory.
>
> - **Selection Logic:** We endeavored to minimize bias by carefully selecting the 18 tasks based on "learnability" and "diversity". The chosen tasks were of moderate difficulty, ensuring the agent could generate successful trajectories for learning. Crucially, they strictly covered all six professional roles defined in the TAC benchmark, guaranteeing a diversity of skills could be acquired.
>
> - **Generalizability of Learned Skills:** To validate the general self-evolution capability, we froze the Memory Module acquired solely from this 18-task subset (~10% of the total tasks). When applied to the full 175-task TAC benchmark, MUSE achieved a new SOTA score of 51.78%, outperforming the previous best by nearly 20%. This outcome strongly demonstrates that the agent learns highly generalizable, transferable knowledge, rather than merely overfitting to the specific learning task subset.

---

> ### Author Response · Authors · 2025-11-25
> **Response to Reviewer SE9S (3/3)**
>
> ## **Q5:** Contextual retrieval relevance & robustness.
>
> **A5:** The specific choice of which experience to retrieve is autonomously determined by the agent. Consequently, we cannot guarantee the absolute relevance of the retrieved content—an inherent challenge that even existing RAG methods cannot fully circumvent.
>
> However, the **Reflect Agent** effectively mitigates this issue. Specifically, if the agent utilizes retrieved experience but fails to complete the task, the Reflect Agent triggers a re-execution of the subtask **without relying on any retrieved information.** Furthermore, it attempts to modify the specific memory entry that may have introduced the noise.
>
>
> ## **Q6:** Comparison across different model capacities
>
> **A6:** We understand your concern regarding the fairness of comparing MUSE (using the generally smaller and less powerful  Flash model) against baselines utilizing stronger models without a direct capacity ablation.
>
> To address this point, and based on your suggestion, we conducted **an additional experiment** by replacing the base model in the MUSE framework with the more capable **Gemini-2.5 Pro**  (using the exact same accumulated experience as the Gemini-2.5 Flash experiment). We tested this configuration on the same $\mathcal{T}_{cl}$ 18-task subset.  The results, shown in Table below, indicate a remarkable performance improvement when the model is upgraded to the stronger Gemini-2.5 Pro under identical experimental conditions.
>
> | Framework |       Model      |  checkpoint | Sckpt (%) ↑ | Avg. Spartial (%) ↑ |
> |:---------:|:----------------:|:-----------:|:-----------:|:-------------------:|
> | OpenHands |  gemini-2.5 pro  |   58 / 85   |    65.39    |        57.67        |
> |    MUSE   | gemini-2.5 flash |  62.2 / 85  |    73.18    |        64.61        |
> |    MUSE   |  gemini-2.5 pro  | **68 / 85** |    **80**   |       **67.1**      |
>
>
> Furthermore, the fact that the MUSE Agent with lighter **Gemini-2.5 Flash** model, outperforms the **OpenHands Agent with Gemini-2.5 Pro**, achieves a cross-model-capacity victory. This strongly validates the efficacy and intrinsic advantages of the MUSE framework design itself.

---

> ### Author Response · Authors · 2025-11-28
> **Gentle Reminder: Feedback Request**
>
> We sincerely appreciate your valuable feedback on the paper. As the deadline for the discussion phase is rapidly approaching, we would be most grateful if you could **share any remaining concerns** at your earliest convenience, allowing us to address them immediately.

---

### Author Response · Authors · 2025-11-25
**Response to All Reviewers**

We sincerely thank all reviewers for their constructive suggestions. We have carefully considered all comments and revised the manuscript as follows:

- **More Precise Definition of the Limitations of Existing Methods:** We now explicitly draw a clear distinction between optimization using reinforcement learning (which typically occurs during the model training phase) and the resource-constrained, high-efficiency "on-the-job" learning scenario during the testing/deployment phase, which is the specific focus of the MUSE framework.
- **Clarification of Memory Utilization in Section 4.4.2:** We have clarified that the memory utilized in the 'DeepSeek-V3 w/ memory' experiment is precisely the experience library accumulated by the MUSE + Gemini-2.5 Flash system after completing three rounds of continuous learning on the task set $\mathcal{T}_{cl}$. This experimental setup provides compelling evidence for our assertion that the experience accumulated by MUSE is both model-agnostic and generalizable.
- **Token Cost and Context Management Analysis in Appendix A.5:** To further validate our approach, we have incorporated new analysis and visual data into the Appendix. This information illustrates the growth trend of token cost with subtask execution, providing clear evidence that our context management mechanism is highly effective at preventing context length explosion and controlling operational costs.

Again, we thank all reviewers for their efforts and time.

Best regards,

The Authors

---

### Meta-Review · Area_Chair_eg8H · 2025-12-16

**Summary:**

This paper proposes MUSE (Plan-Execute-Reflect-Memorize with hierarchical memory: strategic / procedural / tool), a framework that enables LLM agents to self-improve at test-time for long-horizon productivity tasks. It demonstrates high performance on TAC (175 tasks). In the rebuttal, the authors addressed multiple concerns by providing: (i) clarification of positioning relative to existing methods like RL; (ii) analysis of inference time and token/context management; (iii) comparisons under partially identical backend conditions (e.g., OpenHands); (iv) additional experiments on WebArena (comparison with AWM); and (v) analysis of Reflect capability thresholds (performance collapse with weak Reflect) and failure cases.

However, regarding the motivation (the claim regarding the "test-time static" nature of existing models), the positioning relative to related work/novelty, and the fairness of baselines (differences in model capacity and comparison settings), a strong opposing review with high confidence (SE9S) remains. Even after the additional experiments, the evidence is too weak to definitively state that the "concerns have been sufficiently resolved."

Considering the discussion above and the reviewers’ scores, and given the competitive nature of this conference, I regret to recommend rejection of this paper.

**Reviewer Concerns:**

* **Problem Setting / Positioning / Novelty**: "Learning from experience" is already possible via RL and other methods, so the paper's claim (regarding the limitations of "test-time static" models) should be more strictly scoped and organized (SE9S). Furthermore, clarification of the differences and contributions compared to existing trajectory learning and memory-based agents like ExpeL, Memp, and AWM is necessary (aGFx).
* **Baseline Fairness (Model Capacity / Comparison Settings)**: The main TAC comparison involves mixed LLM backends, making interpretation difficult (SE9S, jjy4). While the rebuttal presented additional comparisons with identical model settings, the scope was limited, and uncertainty remains (aGFx’s additional comment reflects this).
* **Generalization (Benchmark Breadth)**: The focus is primarily on TAC; validation on other tasks like OSWorld or SWE-bench is lacking (jjy4, aGFx). The additional WebArena experiment is a step forward, but given the differences in settings (tools, observations, and models), it is still weak support for a strong generalization claim.
* **Operational Cost / Efficiency**: Quantification of inference time, token costs, and training (experience accumulation) costs is important (jjy4, aGFx, st2r). Although time tables and context compression analysis were added in the rebuttal, the overhead for reflection/retrieval remains, and further refinement is desirable to strongly claim practical viability.
* **Robustness of Reflect / Memory**: There are risks involving the accumulation of incorrect memories and dependence on Reflect quality (capability thresholds, maladaptive memories) (aGFx). While additional experiments were beneficial, the systematization of safe operation (verification, cleaning, conflict resolution) remains a future challenge.

**Reviewer Scores:**

* **Reviewer SE9S** (Initial **2**, Conf **5**): Although the authors argued regarding differences from RL (training time/sample efficiency/closed-source constraints) and provided additional experiments, fundamental concerns regarding "motivation/incorporation of related work" and "fairness of comparison" likely remain strong. **2 → 4 (High Confidence).**
* **Reviewer jjy4** (Initial **4**, Conf **2**, First-time): Concerns regarding the single domain, baseline selection, and time/costs have been generally addressed by the WebArena addition and time/token analysis. Although there is no reply, given the original score was a "borderline accept" 4, the primary prediction is **4 → 6 (Low-Mid Confidence)** (though the weight is interpreted as small due to low confidence).
* **Reviewer aGFx** (Initial **4**, Conf **4**): The additional comment explicitly states "mostly satisfied, remaining issues are minor," and major concerns (efficiency, Reflect threshold, failure analysis) have significantly progressed via additional experiments/revisions. **4 → 6 (Mid-High Confidence).**
* **Reviewer st2r** (Initial **6**, Conf **3**): Regarding concerns about learning from failure and inference overhead, evidence of the effects of experience utilization and time comparisons has been presented. Although there is no reply, the stance is unlikely to have changed significantly. **Maintains 6 (Mid Confidence).**

Predicted Average: (4 + 6 + 6 + 6) / 4 = **5.5**.

---

### Decision · Program_Chairs · 2026-01-26

Reject